



# A time-stepping scheme to simulate leaf area index, phenology, and gross primary production across deciduous broadleaf forests in eastern United States

Qinchuan Xin [1], Yongjiu Dai [2], Xiaoping Liu [1]

[1]Guangdong Key Laboratory for Urbanization and Geo-simulation, Sun Yat-sen University, Guangzhou 510275, China
[2]School of Atmospheric Sciences, Sun Yat-sen University, Guangzhou 510275, China

*Correspondence to*: Qinchuan Xin (xinqinchuan@gmail.com); Yongjiu Dai (daiyj6@mail.sysu.edu.cn)

**Abstract.** Terrestrial plants play a key role in regulating the exchange of energy and materials between the land surface and the atmosphere. Robust terrestrial biosphere models that simulate both time series of leaf dynamics and canopy photosynthesis are required to understand the vegetation-climate interactions. This study proposes a time stepping scheme to simulate leaf area index (LAI), phenology, and gross primary production (GPP) simultaneously via only climate variables based on an ecological assumption that plants allocate leaf biomass till an environment could sustain to maximize photosynthetic reproduction. The method establishes a linear function between the steady-state LAI and the corresponding GPP, which is used to track the suitability of environmental conditions for plant photosynthesis, and applies the MOD17 algorithm to form simultaneous equations together, which can be solved numerically. To account for the time lags in plant responses of leaf allocation to environment variation, a time stepping scheme is developed to simulate the LAI time series based on the solved steady-state LAI. The simulated LAI time series is then used to derive the timing of key phenophases and simulate canopy GPP with the MOD17 algorithm. The developed method is applied to deciduous broadleaf forests in eastern United States and has found to perform well on simulating canopy LAI and GPP at the site scale as evaluated using both flux tower and satellite data. The method could also capture the spatiotemporal variation of vegetation LAI and phenology across eastern United States as compared with satellite observations. The developed time-stepping scheme provides a simplified and improved version of our previous modeling approach and forms a potential basis for regional to global applications in future studies.

## 1 Introduction

Terrestrial plants play a key role in regulating the exchange of energy and materials (e.g., radiation, heat and moisture, carbon, and trace gas fluxes) between the land surface and the atmosphere (Beer et al., 2010;Zhu et al., 2017). The canopy structures and characteristics govern solar radiation interception and absorption as well as momentum flows on the land surface (Ni-Meister et al., 2010;Yuan et al., 2013). Individual plants control water transpiration and photosynthetic carbon fixation through processes from transient changes in leaf stomatal conductance to seasonal variation in foliage dynamics (Eagleson, 2005). In turn, external environmental conditions, such as sunlight, temperature, and water and nutrient availability, selectively





determine plant form and function (Bonan, 2008). Numerical terrestrial biosphere model that integrates multidisciplinary knowledges of the Earth science is an essential tool to understand and predict the interactions between terrestrial ecosystems and the climate under a changing global environment.

To match the atmosphere circulation models, developments on the terrestrial biosphere models essentially seek accurate representation as well as the solution to the energy and material exchanging fluxes between ecosystems and the atmosphere. In terrestrial biosphere models, plant canopies are typically characterized using leaf area index (LAI; leaf area per unit ground area) because plant leaf is the basic organ that intercepts solar radiation for photosynthesis and transpiration (Xin et al., 2015b). The exchanging fluxes of energy and materials over vegetation canopy can then be modeled as a function of environmental

conditions (e.g., sunlight, soil moisture, temperature, and humidity) and vegetation LAI (Ding et al., 2014). As the vigorous development of satellite remote sensing technology offers large-scale observations for vegetation monitoring, a number of modeling approaches have been developed to quantify and simulate the land surface fluxes based on climate variables and satellite-derived LAI. These methods, including both the light use efficiency models (e.g., CASA (Potter et al., 1993), the MOD17 algorithm (Running et al., 2004), VPM (Xiao et al., 2004), EC-LUE (Yuan et al., 2010), and TL-LUE (He et al.,

2013)) and the process-based models (e.g., BEPS (Liu et al., 1997), BESS (Ryu et al., 2011), GPD (Xin, 2016), SiB2 (Sellers et al., 1996b)), despite differing from each other on the representation of vegetation processes, have been successfully used for applications from field to global scales. While remote sensing data of vegetation activities perfectly complement the canopy process models, developing the sub-model that could simulate the dynamics of vegetation LAI is fundamental to enhance our abilities on predicting terrestrial ecosystem processes under future scenarios.

Modeling vegetation leaf dynamics via climate variables requires in-depth understanding on plant phenology, of which the modeling is still largely empirical to date and contributes considerable uncertainties to current terrestrial biosphere models (Richardson et al., 2012). One common method for simulating vegetation phenology is to predict the timing of key phenophases such as spring onset and autumn senescence in a growing season (Hufkens et al., 2018;Liu et al., 2018). For example, most

phenology models originate from the Growing Degree Day (GDD) model, a method first proposed by De Réaumur dating back to 1735 (De Réaumur, 1735). The GDD model assumes that plant leaf onset begins when daily mean temperatures accumulated from a fixed date reach a critical threshold. Studies have identified that various environmental factors other than temperature could affect plant phenology to certain degrees (Polgar and Primack, 2011), and therefore, efforts have been made to improve the GDD model by adding different influential factors, such as photoperiod, soil temperature, humidity, and soil

moisture (Melaas et al., 2013;Chuine et al., 1999;Xin et al., 2015a;Yang et al., 2012). Land surface models like the Community Land Model (Oleson et al., 2013) and the Biome-BGC model (White et al., 2000) use a set of complicated and empirical equations to predict the timing of key phenophases across plant functional types. Another method for vegetation phenology modeling is to simulate the entire LAI time series over a growing season. For example, the DeNitrification DeComposition model uses an optimal seasonal growth curve of plant LAI and then calculates environmental stresses of water and nitrogen to





limit daily leaf biomass allocation (Yu et al., 2014). The Growing Season Index as proposed by (Jolly et al., 2005) is a widely used method that could simulate seasonal phenology curves using the climate variables of photoperiod, air temperature, and vapor pressure deficit. While these studies have greatly enriched the availability of the phenology models, the current development of the vegetation phenology models encounters a dilemma: adding different influential climate variables to

account for their impacts on leaf dynamics seems inevitable but certainly complicates the model structure and does not always improve the model accuracies consistently across space and time.

While most of the existing studies choose to develop the plant phenology model independently from the canopy flux model, the physiological processes of leaf distribution and canopy photosynthesis are interrelated. Plants absorb carbon dioxide to

accumulate biomasses through photosynthesis and then redistribute the photosynthetic gain to organs such as leaves, roots, and stems to optimize reproduction. Given limited external resources, plants have evolved to effectively allocate the leaf biomass in response to environment variation so as to maximize photosynthetic carbon gain, the fundamental bioenergy for survival (Givnish, 1986). The strategy of biomass allocation among growth, maintenance, and reproduction in a continuously changing environment directly determines whether plants could persist under natural competition pressures from both inter-

and intra-species (Bonan, 2002). In essence, synthesized analysis of both canopy photosynthesis and leaf distribution processes is needed to solve the difficulties in the development of the current terrestrial biosphere models.

Targeting the current problems associated with vegetation phenology modeling, (Xin, 2016) first proposed a parameterization scheme to simulate vegetation productivity and phenology simultaneously. The method, named as the Growing Production

Day (GPD) model, uses canopy gross primary production (GPP) instead of air temperature as an indicator that synthesizes various environmental factors on plant photosynthesis to track how the environment is suitable for vegetation growth. Analogous to the method that derives potential evapotranspiration, the developed method defines a hypothetic canopy with fixed LAI to model potential GPP (or reference GPP to be precise) under certain environment conditions. Similar to the GDD model, the GPD model predicts vegetation spring onset to occur when the accumulated reference GPP reaches a critical

threshold. The method has been successfully applied to the biomes of evergreen needleleaf forest, deciduous broadleaf forest, and grassland. To allow for predicting the entire LAI time series over a growing season, (Xin et al., 2018) further improved the GPD model by establishing a closed system of equations that includes both vegetation GPP and LAI. The improved GPD model first solves the equations numerically to derive LAI in the steady state and then applies the simple moving average method to obtain the modeled LAI time series. The improved method circumvents the need to empirically prescribe a fixed

canopy and enables modeling of LAI time series in addition to the timing of individual phenophases. There remain shortcomings to overcome for the broad applications of the GPD model. First, the simple moving average method, despite being widely used in many studies, is empirical and does not match with the land surface models that commonly operate at incremental time steps. Second, the developed GPD model that includes many subtle vegetation processes, such as canopy



radiative transfer, leaf stomatal conductance, leaf transpiration, leaf photosynthesis, and soil evaporation, is computationally intensive and requires various climate input data that are often not readily available for regional to global simulations.

Aiming to solve the above-mentioned problems, the objectives of the study are to: 1) develop a time stepping scheme to
simulate leaf dynamics and vegetation productivity, and 2) simplify the GPD model to allow for long-term applications at a large scale. Given that the phenology modeling in deciduous broadleaf forest, a biome that have distinct seasonal growing cycles, still has large uncertainties (Melaas et al., 2016), this study choose to simulate the deciduous broadleaf forests across the eastern United States such that the developed method if successful could provide a potential basis for future applications to other biomes.

## 2 Methods and materials

### 2.1 Modeling steady-state leaf area index

One difficulty in vegetation phenology modeling is that the time scale associated with leaf allocation far exceeds that of many other vegetation processes. Unlike leaf photosynthesis that approaches equilibrium within a minute and stomatal functioning that reaches the steady state in minutes (Sellers et al., 1996a), leaf dynamics takes days or even months in response to climate
variation (Zeng et al., 2013). (Xin et al., 2018) first put forward the concept of the steady-state leaf area index, i.e., canopy LAI when time approaches infinity while the environmental conditions remain unchanging. An alternative biological explanation to the steady-state LAI is the maximum canopy LAI that an environment can sustain infinitely by its own photosynthetic activities. Supposing that the carrying capacity of canopy LAI is proportional to total canopy photosynthetic rate under a given environment, the steady-state LAI can be modeled as follows:

$$LAI_s = mGPP_s \qquad\qquad\qquad\qquad\qquad (1)$$

where $LAI_s$ denotes the steady-state leaf area index; m denotes the constant ratio of steady state leaf area index to environmental capacity; and $GPP_s$ denotes the steady-state gross primary production.

The above equation, despite having a simple form, provides a critical function that complements the canopy photosynthesis model. The only parameter m is dependent on plant functional type and can be quantified from field measurements as the
average ratio of LAI to GPP at canopy closure (i.e., the time when both canopy LAI and GPP reach equilibrium). Studies have developed various canopy photosynthesis models, such as the light use efficiency models and the process-based models. Our previous studies implemented a sophisticated canopy model that assembles the sub-models of canopy radiative transfer, leaf stomatal conductance, leaf transpiration, soil evaporation, and leaf photosynthesis. Although the method has been successfully applied to different biomes, the model structure is complicated for studies at the regional to global scales. To simulate canopy
photosynthesis, this study implements the MOD17 algorithm, a big-leaf light use efficiency model that is used to provide



routine satellite products (Running et al., 2004). The use of the MOD17 algorithm could greatly simplify the modeling processes and reduce the required climate variables, thereby allowing for broad applications. A brief description on the MOD17 algorithm is provided here where details can be found from the user guide of the MODIS GPP product (Running and Zhao, 2015).

Based on the MOD17 algorithm, vegetation GPP can be modeled as follows:

$$\text{GPP}_s = \text{PAR} \times \text{FPAR} \times \varepsilon_{\max} \times f(\text{T}) \times f(\text{VPD}) \tag{2}$$

where $\text{GPP}_s$ denotes the steady-state gross primary production; PAR denotes photosynthetically active radiation; FPAR denotes the fraction of photosynthetically active radiation; $\varepsilon_{\max}$ denotes maximum light use efficiency; and $f(\text{T})$ and $f(\text{VPD})$ denote the scalar functions that account for the limitation of temperature and vapor pressure deficit, respectively, on canopy

10 photosynthesis.

The fraction of photosynthetically active radiation can be modeled as follows (Turner et al., 2006):

$$\text{FPAR} = 1 - \exp(-k\text{LAI}_s) \tag{3}$$

where k denotes the canopy light extinction coefficient and $\text{LAI}_s$ denotes the steady-state leaf area index.

15 The environmental scalars can be modeled as follows:

$$f(\text{T}) = \max\left(\min\left(\frac{\text{TMIN} - \text{TMIN}_{\min}}{\text{TMIN}_{\max} - \text{TMIN}_{\min}}, 1\right), 0\right) \tag{4}$$

$$f(\text{VPD}) = \max\left(\min\left(1 - \frac{\text{VPD} - \text{VPD}_{\min}}{\text{VPD}_{\max} - \text{VPD}_{\min}}, 1\right), 0\right) \tag{5}$$

where TMIN denotes daily minimum air temperature; $\text{TMIN}_{\min}$ and $\text{TMIN}_{\max}$ denote the lower and upper thresholds of daily minimum air temperature for vegetation photosynthetic activities, respectively; VPD denotes daily vapor pressure deficit; and $\text{VPD}_{\min}$ and $\text{VPD}_{\max}$ denote the lower and upper thresholds of daily vapor pressure deficit for vegetation photosynthetic activities, respectively.

Given the environmental conditions, Equations 1 and 2 together form simultaneous equations. Because $\text{LAI}_s$ increases as a linear function of $\text{GPP}_s$ in Equation 1 and $\text{GPP}_s$ increases as a logarithmic function of $\text{LAI}_s$ in Equation 2, the simultaneous equations have one and only one nonzero solution of $\text{LAI}_s$. The nonzero solution can be obtained by implementing a numerical approach that starts with a given initial value of $\text{LAI}_s$ and then solves the equations iteratively until converging.



## 2.2 Modeling leaf area index, phenology, and gross primary production

Because the physiological processes that plants allocate leaf biomass do not respond instantaneously to climate variation, there is a need to simulate vegetation LAI as lagging behind the steady state. One method to account for the time lagging effect is to apply the simple moving average method to buffer abrupt changes from individual events in the time series. Our previous study applied the simple moving average method to model LAI as the unweighted mean of the previous $LAI_s$ as follows (Xin et al., 2018):

$$LAI = \frac{1}{n_{day}} \sum_{i=0}^{n_{day}-1} LAI_s \tag{6}$$

where LAI denotes leaf area index at the n day; $n_{day}$ denotes the number of days; i denotes an index starting from 0 to $n_{day} - 1$; and $LAI_s$ denotes the steady state leaf area index.

The simple moving average method, while showing useful in vegetation phenology modeling, is suitable for retrospective analysis rather than prediction, and importantly, it does not match with most land surface models that operate at incremental time steps. Analogous to the method used to simulate leaf stomatal conductance in response to environmental variation, this study proposes a time stepping scheme to simulate LAI realistically as lagging behind the steady state by a simple restricted growth model (Sellers et al., 1996a) as follows:

$$\frac{dLAI}{dt} = k_l(LAI_s - LAI) \tag{7}$$

where t denotes the time; $k_l$ denotes a time constant that reflects the responses of plant leaf allocation to climate variation; and LAI and $LAI_s$ denote the leaf area index and the steady state leaf area index, respectively.

In the time stepping scheme, vegetation LAI does not change much during winter or summer as the current LAI is close to $LAI_s$, whereas vegetation LAI increases (or decreases) during spring (or autumn) as the current LAI is less (or greater) than $LAI_s$. For example, when the environment turns favorable for plant growth in spring, $LAI_s$ exceeds LAI and dLAI/dt is positive such that the modeled canopy LAI increases. Note that the method developed here essentially uses the canopy photosynthetic capacity (i.e., the steady-state gross primary production) instead of air temperature as a synthesized indicator to track the suitability of the environment to plant growth in time series, and therefore, the developed method is referred to as the Simplied Growing Production-Day (SGPD) model following our previous studies (Xin et al., 2018).

Given the modeled LAI time series, both vegetation phenology and canopy GPP can be easily retrieved. Various approaches have already been developed to derive the timing of key phenophases such as spring onset and autumn senescence from seasonal LAI trajectories. This study models the phenological transition dates using a simple method that derives the first



spring and last autumn dates at which LAI reaches 20%, 50%, and 80% of the seasonal amplitudes (Richardson et al., 2012). The selected relative amplitudes (20%, 50%, and 80%) are correspondent to different plant growth stages over a growing season. Given temperature and vapor pressure deficit, the canopy GPP is simply modeled by substituting the modeled LAI time series back into the MOD17 algorithm.

## 2.3 Comparative studies using Growing Season Index

The Growing Season Index (GSI), a widely used method in vegetation phenology modeling, allows for modeling seasonal LAI time series rather than individual phenophases and is implemented to make direct comparisons with the SGPD model (Jolly et al., 2005). The GSI model performs comparably to or even outperforms other terrestrial biosphere models on predicting the timing of key phenophases for deciduous broadleaf forests (Melaas et al., 2013).

The instantaneous GSI is first derived based on the work of (Jolly et al., 2005) as follows:

$$\text{iGSI} = \text{iTMIN} \times \text{iVPD} \times \text{iPhoto} \tag{8}$$

where iGSI denotes instantaneous growing season index; and iTMIN, iVPD, and iPhoto denote the instantaneous scalar functions that account for the constraints of daily minimum air temperature, vapor pressure deficit, and photoperiod, respectively, on vegetation growth. The scalar functions for iTMIN, iVPD, and iPhoto have the mathematic forms similar to Equations 4 and 5 and are derived the same as defined in (Jolly et al., 2005).

LAI can be modeled as the simple moving average of the instantaneous GSI scaled using maximum LAI as follows:

$$\text{GSI} = \frac{1}{n_{\text{day}}} \sum_{i=0}^{n_{\text{day}}-1} \text{iGSI} \tag{9}$$

$$\text{LAI} = \text{GSI} \times \text{LAI}_{\text{max}} \tag{10}$$

where GSI denotes growing season index at the n day; $n_{\text{day}}$ denotes the number of days; i denotes an index starting from 0 to the previous one day; iGSI denotes the instantaneous growing season index; and $\text{LAI}_{\text{max}}$ denotes the maximum leaf area index at canopy closure.

It is noteworthy that the instantaneous GSI uses the product of the scalars of minimum temperature, vapor pressure deficit, and photoperiod as an indicator to track the potential canopy photosynthetic capacities on the daily basis. Both the GSI model and the SGPD model, despite having different forms, share the same modeling idea. To understand the differences between the simple moving average method and the time stepping method, the GSI model is also implemented with the simple restricted growth model as follows:



$$LAI_s = iGSI \times LAI_{max} \tag{11}$$

$$\frac{dLAI}{dt} = k_l(LAI_s - LAI) \tag{12}$$

where iGSI denotes the instantaneous growing season index; $LAI_{max}$ denotes the maximum leaf area index at canopy closure; $k_l$ denotes a time constant that accounts for the lagged responses of plant leaf allocation to climate variation; and LAI and $LAI_s$ denote the leaf area index and the steady state leaf area index, respectively.

With the modeled LAI time series, the phenological transition dates are then retrieved based on the seasonal amplitude ratio method, the same way as processing the LAI time series derived from the SGPD model. Vegetation GPP is modeled by substituting the modeled LAI time series into the MOD17 algorithm.

**2.4 Study materials**

Our modeling studies are made at the site scale using both the flux tower data and remote sensing data and at the regional scale
using both the climate data and remote sensing data for deciduous broadleaf forests in eastern United States. In the site-scale studies, all the flux tower sites of deciduous broadleaf forests (Table 1) that are available in the AmeriFlux website (http://ameriflux.ornl.gov/) were used for analysis. As the developed SGPD model is a simplified version of our previous modeling approach, the site-scale modeling studies only require daily incoming solar radiation, minimum air temperature, vapor pressure deficit, photoperiod, LAI, and GPP data. Daily incoming solar radiation, vapor pressure deficit, and GPP have
already been provided in the Level 4 flux tower data, whereas daily minimum air temperature was processed from the half-hourly gap-filled Level 2 data and daily photoperiod as required by the GSI model was computed based on Equation 13 as a function of geolocation and the day of year (Allen et al., 1998). As the MODIS LAI has been found to match field measurements well for deciduous broadleaf forests in eastern United States (Myneni et al., 2002), the 8-day 500 m MODIS LAI Version 6 products (MOD15A2H) that are downloaded from the Land Processes Distributed Active Archive Center
(https://lpdaac.usgs.gov/) were used as the reference data. Canopy LAI at each site were extracted from MOD15A2H for the pixel that contains the corresponding site. The extracted 8-day MODIS LAI if identified as poor quality in MOD15A2H were replaced using the three-point median-value moving window technique. Spikes in the LAI time series were removed using the Hampel filter and then gap-filled using the autoregressive modeling approach (Akaike, 1969). The obtained 8-day LAI time series were further smoothed using the Savitzky-Golay filter and then linearly interpolated to generate daily time series. The
phenological transition dates were retrieved from daily LAI time series using the method that derives the first spring and last autumn dates at which LAI reaches 20%, 50%, and 80% of the seasonal amplitudes, respectively (Richardson et al., 2012).

$$Pho = \frac{24}{\pi} \arccos\left(-\tan(\varphi)\tan\left(0.409\sin\left(\frac{2\pi}{365}DOY - 1.39\right)\right)\right) \tag{13}$$

where Pho denotes daily photoperiod; $\varphi$ denotes the latitude; and DOY denotes the day of year.



**Table 1: Site information for the studied flux towers of deciduous broadleaf forests.**

| Site Code | Site Name | Lat (°N) | Lon (°W) | Elev (m) | Years | Reference |
|---|---|---|---|---|---|---|
| US-Bar | Bartlett Experimental Forest | 44.0646 | -71.2881 | 272 | 2004-2011 | Jenkins et al. (2007) |
| US-ChR | Chestnut Ridge | 35.9311 | -84.3324 | 286 | 2006-2010 | Hollinger et al. (2010) |
| US-Dk2 | Duke Forest Hardwoods | 35.9736 | -79.1004 | 168 | 2007-2008 | Oishi et al. (2008) |
| US-Ha1 | Harvard Forest EMS Tower | 42.5378 | -72.1715 | 340 | 2000-2012 | Urbanski et al. (2007) |
| US-MMS | Morgan Monroe State Forest | 39.3231 | -86.4131 | 275 | 2000-2014 | Dragoni et al. (2011) |
| US-MOz | Missouri Ozark | 38.7441 | -92.2000 | 219 | 2005-2013 | Gu et al. (2006) |
| US-Oho | Oak Openings | 41.5545 | -83.8438 | 230 | 2005-2011 | Xie et al. (2014) |
| US-Slt | Silas Little Experimental Forest | 39.9138 | -74.5960 | 30 | 2005-2012 | Clark et al. (2012) |
| US-UMB | Univ. of Mich. Biological Station | 45.5598 | -84.7138 | 234 | 2000-2012 | Gough et al. (2013) |
| US-UMd | UMBS Disturbance | 45.5625 | -84.6975 | 239 | 2008-2012 | Gough et al. (2013) |
| US-WBW | Walker Branch | 35.9588 | -84.2874 | 343 | 2000-2006 | Miller et al. (2007) |
| US-WCr | Willow Creek | 45.8060 | -90.0798 | 515 | 2000-2013 | Desai et al. (2008) |

Our regional-scale studies used both the climate data and satellite remote sensing data from 1982 to 2016. The daily 1000 m
Daymet Version 3 dataset (Thornton et al., 2012) was downloaded from the Oak Ridge National Laboratory Distributed Active
Archive Center (http://daymet.ornl.gov/). The Daymet dataset provided daily incoming solar radiation, minimum temperature,
vapor pressure, and photoperiod data and we derived daily vapor pressure deficit as the difference between average saturated
vapor pressure and vapor pressure. Two different satellite LAI products, including the Global Land Surface Satellite (GLASS)
dataset (Xiao et al., 2014) spanning from 1982 to 2014 and the MODIS LAI dataset (Myneni et al., 2002) spanning from 2001
to 2016, were used for the regional studies. The 8-day GLASS LAI product was generated at the 0.05° resolution using the
AVHRR data for the time period from 1982 to 1999 and at the 1000 m resolution using the MODIS data for the time period
from 2000 to 2012. The 8-day satellite LAI data across eastern United States were processed the same way as the processing
of the site-scale data to obtain daily LAI time series. Because seasonal LAI amplitudes for each individual pixel could vary
from year to year, the 2001-2010 average seasonal LAI amplitude were used as a baseline to derive the start of the season
(SOS) and the end of the season (EOS) for each pixel for each year as the dates when seasonal LAI reaches 50% of the multi-
year average seasonal LAI amplitude. The growing season length (GSL) was derived as the difference between EOS and SOS.
A 500 m MODIS-based land cover map was obtained from the USGS Land Cover Institute (https://landcover.usgs.gov/). The
land cover map was generated by choosing the land cover classification with the highest overall confidence for each pixel in
10-year (2001-2010) Collection 5.1 MODIS land cover type (MCD12Q1) data (Broxton et al., 2014). The 500 m land cover
map was resampled to 1000 m resolution using the majority resampling approach and was reprojected to the Lambert
Conformal Conic projection to mask areas that are not covered by deciduous broadleaf forests.



## 2.5 Model parameterization and comparison

As the MOD17 algorithm is a well-parameterized model, this study applies the model parameters in literature directly. Following the user guide of the MODIS GPP product (Running and Zhao, 2015), key parameters in the MOD17 algorithm are set as $\varepsilon_{max} = 1.165$ gC/MJ , $TMIN_{min} = -6.0\ °C$, $TMIN_{max} = 9.94\ °C$, $VPD_{min} = 0.65$ kPa, and $VPD_{max} = 1.65$ kPa. The light extinction coefficient of the canopy is 0.5. The parameter that defines the ratio of leaf area index to environmental capacity is set as m $= 0.58$ m²(leaf area)/gC/day as quantified using the average ratio of LAI to GPP at canopy closure using the flux tower data. The canopy maximum LAI is set as 5.80 based on the maximum 95th percentile of satellite-derived LAI across sites and years (Xin et al., 2018). The parameter $n_{day}$ in the simple moving average method and the parameter $k_l$ in the time stepping method control the response of plant leaf allocation to environmental variation. The parameter $n_{day}$ is set as 21 days and the parameter $k_l$ is calibrated as 0.080 day⁻¹.

This study compares four different modeling approaches, including the results simulated using both the SGPD model and the simple moving average method (hereinafter referred to as SGPD-SMA), using both the SGPD model and the time stepping scheme (hereinafter referred to as SGPD-TS), using both the GSI model and the simple moving average method (hereinafter referred to as GSI-SMA), and using both the GSI model and the time stepping scheme (hereinafter referred to as GSI-TS). The commonly used metrics, including the Pearson correlation coefficient (R), the coefficient of determination (R²), the root-mean-square error (RMSE), and the mean bias error (MBE), are derived for model assessment and comparison.

## 3 Results

### 3.1 Site-scale modeling

Figure 1 shows an example for the simulated time series of LAI and GPP using data acquired at the US-WCr site in 2010. The LAI time series simulated using both the SGPD-SMA and SGPD-TS methods are consistent with that obtained from MODIS. The LAI simulated using both the GSI-SMA and GSI-TS methods could also capture the observed seasonal variation of LAI but the modeled phenophases obviously have a leading phase in spring and a lagging phase in autumn as compared with observations. For both the SGPD model and the GSI model, the results derived using the time stepping method are consistent with those derived using the simple moving average method, indicating that the time stepping method is an effective way to reflect the lagging responses of plant leaf allocation to environmental conditions. By substituting the time series of LAI derived from different modeling approaches into the MOD17 algorithm, all the simulated GPP time series could match the flux tower measurements. Daily fluctuation in the observed GPP time series is largely due to variation in solar radiation from day to day and is well captured by the models. The GPP modeled using both the GSI-SMA and GSI-TS methods have slight overestimates in the phenological transition periods like spring and autumn and match well with the flux tower observations in summer and winter.



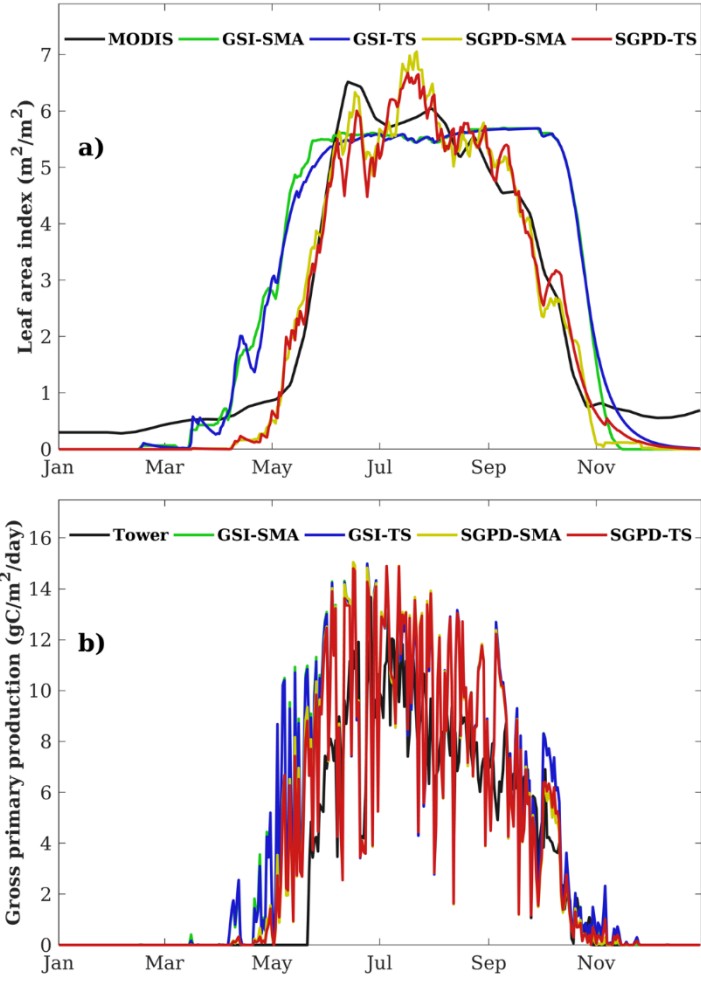

**Figure 1: The modeled and measured daily time series of a) leaf area index and b) gross primary production are shown for the flux tower site of US-WCr in 2010. The reference LAI time series in Figure 1a are derived from the MODIS data and the reference GPP time series in Figure 1b are obtained from the flux tower measurements.**

Figure 2 shows the regression analysis between the modeled and satellite-derived LAI. Overall, the SGPD model outperforms the GSI model on modeling LAI. When evaluated against the MODIS LAI data, the SGPD-SMA and SGPD-TS models achieved the R² of 0.887 and 0.890, respectively, and the RMSE of 0.804 and 0.778 m²/m², respectively, whereas the GSI-SMA and GSI-TS models achieved the R² of 0.746 and 0.759, respectively, and the RMSE of 1.356 and 1.303 m²/m², respectively. Both the GSI-SMA and GSI-TS models simulate LAI reasonably in summer and winter but overestimate LAI in spring and autumn, and therefore, the strong correlations between the GSI-modeled and MODIS-derived LAI are largely due to the underlying seasonality of deciduous broadleaf forests. It is noteworthy that the time stepping method and the simple





moving average method, despite having different mathematical expressions, generate nearly the same simulation results. The R² values between the SGPD-TS model and the SGPD-SMA model and between the GSI-TS model and the GSI-SMA model are 0.989 and 0.994, respectively, and the regression lines are close to the lines of equity, indicating that the time stepping method is an alternative representation for the simple moving average method. Because ecosystem processes are typically

5    simulated at incremental time steps in terrestrial biosphere models, the time stepping method undoubtedly is more suitable than the simple moving average method for future uses in the land surface models.



**Figure 2: Scatter plots are shown for the comparisons a) between the SGPD-SMA LAI and the MODIS LAI, b) between the SGPD-**
10   **TS LAI and the MODIS LAI, c) between the SGPD-TS LAI and the SGPD-SMA LAI, d) between the GSI-SMA LAI and the MODIS LAI, e) between the GSI-TS LAI and the MODIS LAI, and f) between the GSI-TS LAI and the GSI-SMA LAI on a weekly basis. All available site-year flux tower data were included in the analysis. The solid lines denote the 1:1 lines and the dashed lines denote the regression lines.**

15   Table 2 lists the statstitcal metrics that illustrate the model performance on prediting the timing of different phenophases are listed in. The use of phenophases in model assessment eliminates the seasonality effects of the LAI times series. As evaluated





against satellite observations, the SGPD-SMA model could well retrieve the spring onset dates when LAI reaches 50% seasonal amplitude and the obtained correlation coefficient is 0.718 with RMSE of 13.04 days. The SGPD-TS model performs comparable to the SGPD-SMA model and the resulted corretion coefficients are all significant expect for the dates that autumn LAI reaches 80% seasonal amplitudes. The SGPD-based models generally outperforms the GSI-based models as the achieved

correlation coefficients are higher and the RMSE are smaller for more than 10 days. Both the GSI-SMA and GSI-TS models predict spring onsets earlier than observations for more than 30 days and predict autumn senenssence later than observations for more than 20 days. By comparison, the SGPD-TS model predicts the dates that spring and autumn LAI reaches 50% seasonal amplitudes well with the MBE of only -2.56 and -2.86 days, respectively.

**Table 2: The performance of the modeled timings of phenophases as evaluated against satellite observations. The timings of phenophases were derived based on dates at which the leaf area index reaches 20%, 50%, 80% of seasonal amplitude. Positive mean bias error (MBE) indicates that the modeled spring onsets are earlier than the observed ones and negative MBE indicates the opposite.**

| phenophases | SGPD-SMA | | | SGPD-TS | | | GSI-SMA | | | GSI-TS | | |
|---|---|---|---|---|---|---|---|---|---|---|---|---|
| | R | RMSE | MBE | R | RMSE | MBE | R | RMSE | MBE | R | RMSE | MBE |
| | | (days) | (days) | | (days) | (days) | | (days) | (days) | | (days) | (days) |
| Spring LAI 20% | 0.790*** | 16.17 | -10.85 | 0.824*** | 13.37 | -8.34 | 0.763*** | 40.38 | -38.30 | 0.770*** | 39.62 | -37.58 |
| Spring LAI 50% | 0.718*** | 13.04 | -1.97 | 0.691*** | 13.68 | -2.56 | 0.653*** | 38.47 | -34.92 | 0.657*** | 38.22 | -34.63 |
| Spring LAI 80% | 0.432*** | 20.91 | 12.63 | 0.409*** | 21.19 | 12.41 | 0.560*** | 32.86 | -28.00 | 0.565*** | 28.54 | -23.55 |
| Autumn LAI 80% | 0.220 | 31.80 | -25.56 | 0.164 | 27.90 | -20.64 | 0.021 | 35.38 | 32.42 | -0.004 | 35.23 | 32.27 |
| Autumn LAI 50% | 0.686*** | 9.80 | -5.42 | 0.625*** | 9.48 | -2.86 | 0.621*** | 24.20 | 23.07 | 0.616*** | 24.63 | 23.51 |
| Autumn LAI 20% | 0.703*** | 8.87 | 2.15 | 0.676*** | 10.91 | 6.37 | 0.689*** | 19.64 | 18.48 | 0.713*** | 22.93 | 22.00 |

The modeled and measured GPP are compared in Figure 3 to address the key question that whether the simulated LAI could be applied to model canopy GPP. Compared with the flux tower measurements, the results modeled using the SGPD-SMA, SGPD-TS, GSI-SMA, and GSI-TS LAI could achieve the R² values of 0.768, 0.773, 0.722, and 0.719, respectively, and the RMSE values of 2.273, 2.239, 2.577, 2.535 gC/m²/day, respectively. The modeled results using the GSI-based LAI have higher errors, in terms of both RMSE and MBE, than those using the SGPD-based LAI. The accuracies of the modeled GPP using

the SGPD-based LAI are only slightly lower than to that using the MODIS-based LAI directly. The modeling results obtained based on the simple moving average method are nearly the same as those obtained based on the time stepping method. Given the high degrees of consistency between the simple moving average method and the time stepping method on modeling LAI, phenology, and GPP, only the results obtained using the time stepping method are shown and discussed in the regional studies as presented in the following section.



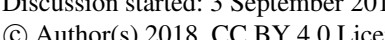


**Figure 3: Scatter plots are shown for the comparisons a) between the GPP modeled using SGPD-SMA LAI and the flux tower GPP, b) between the GPP modeled using SGPD-TS LAI and the flux tower GPP, c) between the GPP modeled using MODIS LAI and the flux tower GPP, d) between the GPP modeled using GSI-SMA LAI and the flux tower GPP, and e) between the GPP modeled using GSI-TS LAI and the flux tower GPP on a weekly basis. All available site-year flux tower data were included in the analysis. All the modeled GPP were derived using the MOD17 algorithm. The solid lines denote the 1:1 lines and the dashed lines denote the regression lines.**

## 3.2 Regional-scale modeling

Figure 4 shows the spatial extents of the 10-year (2001-2010) mean LAI and associated errors as derived from remote sensing data and model simulations. The SGPD-TS method could well capture the spatial pattern of the satellite-derived LAI, including the decreasing gradients from south to north and the decreases in mountain areas (Figure 4a and 4b). The 10-year mean LAI derived from the GSI-TS method (Figure 4c) also show a decreasing trend from south to north but the modeled LAI is much larger than the MODIS LAI. Because the GSI-TS method defines the maximum leaf area index for the growing season, the





overestimation on the modeled 10-year mean LAI is primarily due to model overestimates in the spring and autumn phenological transitions. As compared with the MODIS observations, RMSE and MBE obtained by the SGPD-TS method are much smaller than and distribute more evenly than those obtained by the GSI-TS method. RMSE for the GSI-TS LAI exhibit a decreasing north-south gradient, implying that the model accuracies are lower in southern areas lower than in northern areas.

5   MBE for the GSI-TS model are greater than 0.5 m²/m² for most areas. When comparing SGPD-TS LAI with MODIS LAI, RMSE are less than 0.5 m²/m² and MBE are minor across the study region. The amplitudes of the error metrics in the regional-scale studies are consistent with those in the site-scale studies. Note that some studies applied the multi-year mean LAI as derived from the remote sensing data to simulate the land surface processes, the results obtained here indicate that the SGPD-TS method can be used alternatively to provide multi-year mean LAI time series via climate variables for land surface studies.





**Figure 4: The spatial extents are shown for a) the 2001-2010 mean MODIS LAI, b) the 2001-2010 mean SGPD-TS LAI, c) the 2001-2010 mean GSI-TS LAI, d) RMSE between SGPD-TS LAI and MODIS LAI, e) RMSE between GSI-TS LAI and MODIS LAI, f) MBE between SGPD-TS LAI and MODIS LAI, and g) MBE between GSI-TS LAI and MODIS LAI across eastern United States. The units for both RMSE and MBE are m² (leaf area) per m² (ground area).**

The spatial extents for the 10-year mean phenological metrics including the start of the season (SOS), the end of the season (EOS), and the growing season length (GSL) are shown in Figure 5. The SGPD-TS method predicts lower SOS (i.e., earlier spring onset), higher EOS (i.e., later autumn senescence), and longer GSL in southern areas than in northern areas. The spatial



distributions of all phenological metrics derived using SGPD-TS LAI agree well with those derived using MODIS LAI. From the statistical analysis as shown in the subplots, the phenological metrics derived from the SGPD-TS method could achieve the correlation coefficient values of 0.879, 0.552, and 0.844, the RMSE values of 8.13, 7.54, and 13.73 days, and the MBE values of 0.71, -2.82, and -3.54 days, for SOS, EOS, and GSL, respectively, as compared to those derived from the MODIS

5   data. Although the spatial distributions of the phenological metrics derived from the GSI-TS method match those derived from the satellite observations, the modeled results have considerable biases, where the RMSE values are 38.05, 14.37, and 51.58 days, and the MBE values are -36.33, 12.91, and 49.23 days, for SOS, EOS, and GSL, respectively. Consistent with the site-scale studies, the GSI-TS method predicts spring onset much earlier and autumn senescence later than the satellite-derived data, resulting in large overestimation of the growing season length. Despite having limited model accuracies as compared

10  with observations, the GSI-TS method here performs comparable to or even better than the other phenology models in similar modeling studies.





**Figure 5: The spatial extents are shown for a) the start of the season (SOS) derived from MODIS LAI, b) SOS derived from SGPD-TS LAI, c) SOS derived from GSI-TS LAI, d) the end of the season (EOS) derived from MODIS LAI, e) EOS derived from SGPD-TS LAI, f) EOS derived from GSI-TS LAI, g) the growing season length (GSL) derived from MODIS LAI, h) GSL derived from SGPD-TS LAI, and i) GSL derived from GSI-TS LAI using the 10-year (2001-2010) mean data across eastern United States. The embedded subplots show the comparisons between modeled and MODIS-derived phenological metrics for SOS, EOS, and GSL, respectively.**



Figure 6 displays the multi-year phenology anomalies that are spatially averaged for deciduous broadleaf forest across eastern United States. The use of phenology anomalies relative to the 2001-2010 average instead of absolute values makes the results directly comparable. The SGPD-TS method could capture the interannual variation of vegetation phenology retrieved from the remote sensing data. When comparing the SGPD-TS method with the MODIS (2001-2016) data, the correlation

5    coefficients are 0.896 (p<0.001), 0.650 (p=0.006), and 0.817 (p<0.001), for SOS, EOS, and GSL, respectively. When comparing the SGPD-TS method with the GLASS (1982-2014) data, as derived from and the correlation coefficients are 0.554 (p=0.001), 0.717 (p<0.001), 0.637 (p<0.001), for SOS, EOS, and GSL, respectively. The SGPD-TS method outperforms the GSI-TS method on capturing the long-term trends of vegetation phenophases, as the correlation coefficients obtained using the GSI-TS method are lower and sometimes insignificant. Yearly fluctuation in EOS derived using the GSI-TS method is

10   smaller than those derived from both the SGPD-TS method and the satellite data. The SOS and EOS derived from the GLASS data have much larger variation in 1982-2000 than in 2001-2010, suggesting that the use of the AVHRR and MODIS data in the GLASS dataset could contribute uncertainties in the satellite-derived phenological metrics. Both Figure 5 and 6 indicate that the SGPD-TS method is reliable on capturing the spatiotemporal patterns of regional vegetation phenophases.





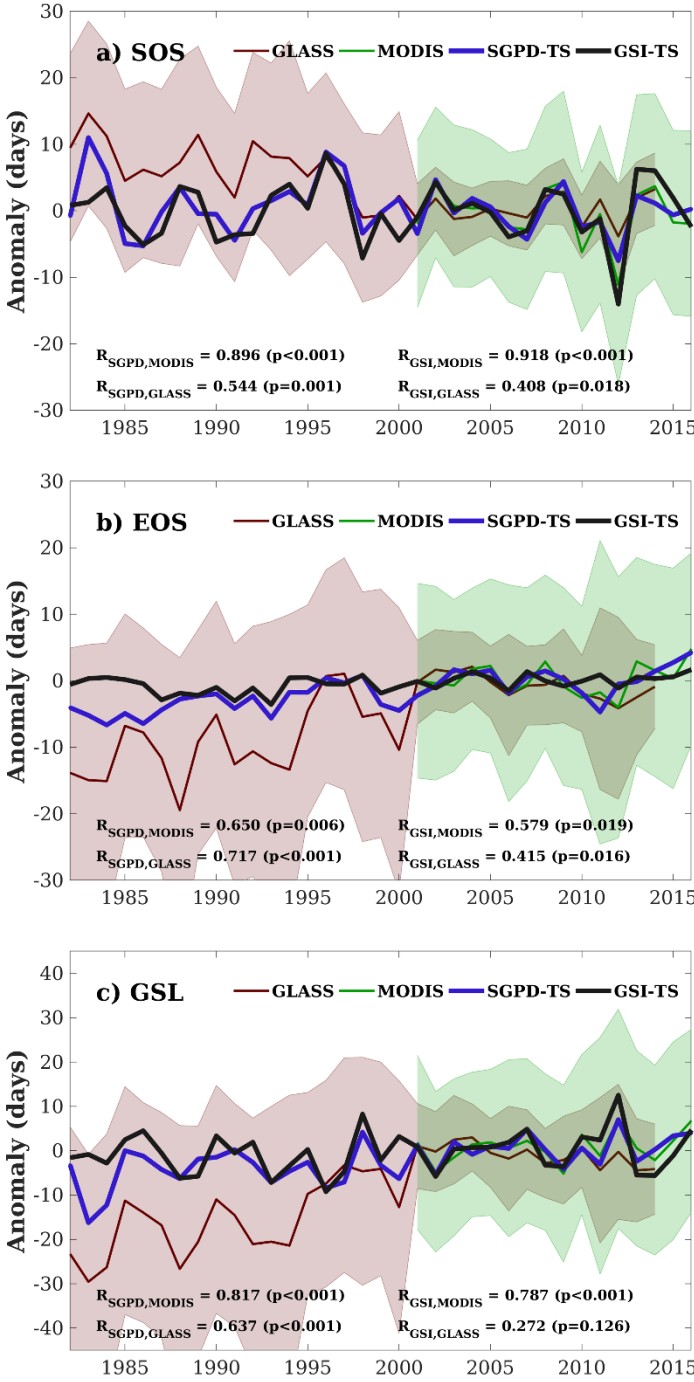

**Figure 6: The spatially-averaged phenology anomalies relative to the 2001-2010 average are shown for a) the start of the season (SOS), b) the end of the season (EOS), and c) the growing season length (GSL). SOS and EOS are derived as the date that LAI first and last reaches 50% of the seasonal amplitudes and GSL is derived as the difference between EOS and SOS.**





Figure 7 compares the simulated c using the MOD17 algorithm and LAI derived from different approaches. The 10-year average annual GPP obtained using SGPD-TS LAI has a similar spatial pattern with that obtained using MODIS LAI and apparently has lower values than that obtained using GSI-TS LAI. Taking the GPP simulated using MODIS LAI as reference,

5   the results simulated using SGPD-TS LAI achieve the correlation coefficient of 0.898 with RMSE of 78.78 gC/m²/year and MBE of 12.22 gC/m²/year, whereas the results simulated using GSI-TS LAI achieve the correlation coefficient of 0.898 with RMSE of 173.45 gC/m²/year and MBE of 153.43 gC/m²/year. Although the obtained correlation coefficients are close, the SGPD-TS method results in the regression lines closer to the 1:1 lines with smaller bias errors than the GSI-TS method. The zonally average profiles of the 2001-2010 average annual GPP as shown in Figure 7d suggest that the results obtained from

10   the SGPD-TS method are close to those obtained using MODIS LAI, whereas the results obtained from the GSI-TS method have positive biases of approximately 120 - 180 gC/m2/year (roughly 10 - 15%) across latitudes. Note that the MOD17 algorithm has positive MBE of 0.247 gC/m2/day and 0.571 gC/m2/day when using MODIS LAI and GSI-TS LAI, respectively, as model input data in the site-scale study. The differences in MBE between the two modeling methods are 0.324 gC/m2/day (or 118.26 gC/m2/year in equivalence) for the site-scale studies, which are consistent with the regional-scale studies.

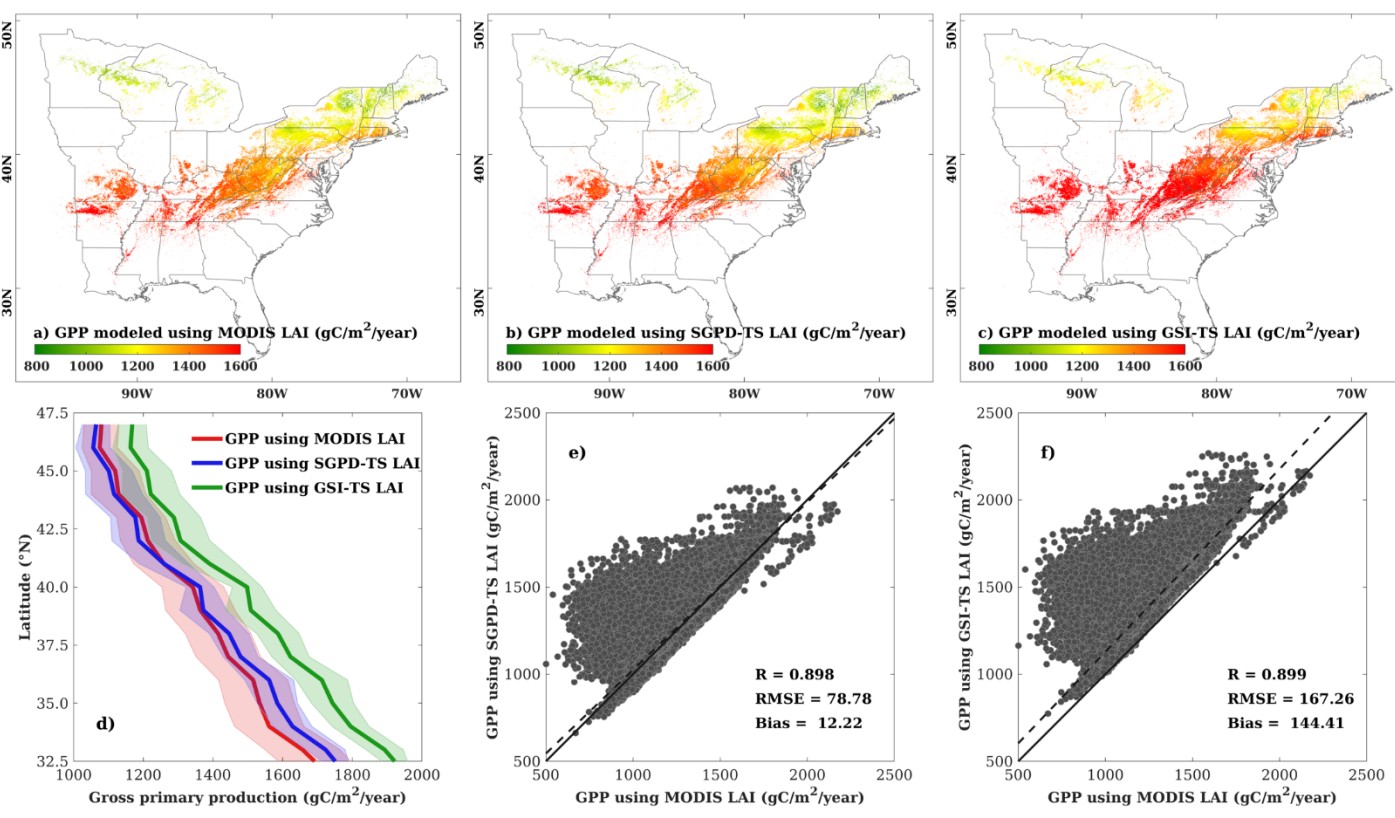



**Figure 7: Comparisons are shown for a) the spatial extent of annual GPP modeled using MODIS LAI, b) the spatial extent of annual GPP modeled using SGPD-TS LAI, c) the spatial extent of annual GPP modeled using GSI-TS LAI, d) the zonally averaged profiles of annual gross primary production modeled using LAI derived from different approaches, e) the regression between GPP modeled using SGPD-TS LAI and MODIS LAI, and f) the regression between GPP modeled using GSI-TS LAI and MODIS LAI. The**
**simulated daily GPP were first summed for each individual year and were then averaged across years to derive the 2001-2010 average annual GPP as shown in Figure 7a, 7b, and 7c. The shaded areas in Figure 7d mark the range of the standard deviation. All pixels of deciduous broadleaf forest across eastern United States are included in analysis in Figure 7e and 7f.**

## 4 Discussion

Here we provide a solution that bridges the canopy photosynthesis model and the leaf dynamics model, which overcomes the
weakness that existing studies developed the plant phenology model independent of the canopy photosynthesis model. The developed method first proposes a linear function between the canopy photosynthetic capacity and the steady state LAI so as to complement the canopy photosynthesis model and then applies a simple restricted growth model to account for the lagged responses of plant leaf allocation to natural environment. In essence, the developed method, although having a simple form, has synthesized the impacts of various climate factors on leaf dynamics because any climate variable that influences vegetation
photosynthesis would affect the process of plant leaf allocation in the models as well. Consistent with field observations, the simulated LAI increases as the environmental conditions turn favorable for photosynthetic activities such as increases in photoperiod and temperature. It is worth noting that our modeling approach is essentially based on an ecological assumption that plants have evolved strategies to optimize leaf distribution according to the environmental carrying capacity for maximizing photosynthetic carbon gain. The assumption undoubtedly requires observational supports from extensive
experiments, preferably laboratory controlling experiments, on plant physiology and phenology in future studies.

The performance of our developed method is largely dependent on the canopy photosynthesis model used. In our previous studies, we developed a process-based canopy photosynthesis model that synthesizes sub-models such as canopy radiative transfer, leaf transpiration, leaf stomatal conductance, leaf photosynthesis, and soil evaporation and applied it for modeling the
LAI time series. When applying the simple moving average method, implementing the process-based model in Xin et al. (2018) achieved higher accuracies than implementing the MOD17 algorithm on modeling canopy GPP and LAI as reflected by higher R² and lower errors. It implies that the LAI modeling in our developed method likely benefits from improvements on the canopy photosynthesis model. This study chooses the MOD17 algorithm instead of the sophisticated process-based model because the MOD17 algorithm is well parameterized across biomes and requires quite limited model inputs of climate variables.
Successful implementation with the MOD17 algorithm allows for extending the developed method to applications across biomes at regional to global scales.

The time stepping scheme developed here is also an improvement over the simple moving average method as used in previous studies. The results obtained using the time stepping method are consistent with the simple moving average method at the site
scale and show to be reasonable at the regional scale. Compared to the simple moving average method, the time stepping





method could fit seamlessly into the land surface models that operate at incremental time steps such as the Community Land Model and the Common Land Model (Dai et al., 2003). Because the state-of-the-art land surface models all include the canopy photosynthesis sub-model, the developed method can then be easily embedded into these land surface models as an alternative phenology model. Compared to the simple light use efficiency model like the MOD17 algorithm, implementation of the

developed time-stepping scheme in the land surface models relies on supercomputing for global applications. To better understand the performance of the develop method, one study is now undertaken to implement the developed method with the Common Land Model for simulating multi-decadal LAI and GPP for global biomes via only climate variables.

Applying the developed method to other biomes and other regions still has issues to be solved appropriately. The time stepping

method uses the parameter $k_l$ to account for the time lags of leaf allocation in response to environmental changes. For the deciduous broadleaf forests, a biome with strong seasonality, the developed scheme achieved reasonable results with appropriate parameterization. Short vegetation like grasslands tends to respond much quickly to abrupt environment changes like precipitation and tropical ecosystems have strong resilience to short-term environmental variation (Shen et al., 2011;Levine et al., 2016). These understandings from the observational studies imply that biomes have varied response speeds

to the environment and proper model calibration and assessment are required for the developed method. Using the observation data from remote sensing alone is inadequate for model development as satellite-derived LAI could have large uncertainties for some specific biomes other than deciduous broadleaf forests. Fortunately, global flux tower network and regional phenology observation networks are now established and offer abundant data for comprehensive model assessment.

## 5 Conclusions

Robust terrestrial biosphere model is a basic tool for understanding the interactions between the land surface and the atmosphere. To provide a complete solution to the simulation of plant leaf dynamics and canopy photosynthesis, this study establishes a linear relationship between the steady state leaf area index and the corresponding canopy photosynthetic capacity based on the idea that plants optimize leaf biomass allocation according to the environmental carrying capacity for maximizing photosynthetic carbon gain. The proposed leaf allocation function complements the canopy photosynthesis model of the

MOD17 algorithm to form simultaneous equations that can be solved iteratively. To account for the time lagging of plant leaf allocation in response to climate variation, a time stepping scheme based on a simple restricted growth model is applied to the solved steady state leaf area index to obtain time series of leaf area index. The developed method could perform reasonably well on simulating leaf area index, phenology, and gross primary production for deciduous broadleaf forests across eastern United States over years as found in both the site-scale and regional-scale modeling studies. Compared to the simple moving

average method, the time stepping scheme developed here is consistent with and can be easily embedded into the state-of-the-art land surface models that typically operate at incremental time steps. The developed method allows for simulating leaf area





index and gross primary production simultaneously and provides a much simplified and improved version of our previous model as a basis for global applications in future studies.

**Author contributions**

Qinchuan Xin designed the experiments and performed the simulations. All authors interpreted the results. Qinchuan Xin wrote the manuscript with contributions from all coauthors. The authors declare that they have no conflict of interest.

**Acknowledgments**

We thank the researchers and investigators who are involved in collecting and sharing the AmeriFlux dataset. This research is supported by National Key R&D Program of China (grant nos. 2017YFA0604302 and 2017YFA0604402). We also thank anonymous reviewers for their constructive comments.

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
