# Peer review of "A simple time-stepping scheme to simulate leaf area index, phenology, and gross primary production across deciduous broadleaf forests in eastern United States"

_Biogeosciences, 2018_

## Referee Comment (RC1) · Anonymous Referee #1 · 27 Sep 2018

The study provided by Q. Xin et al "A time-stepping scheme to simulate leaf area index, phenology, and gross primary production across deciduous broadleaf forests in eastern United States" is mainly focused on development of a new modeling algorithm to parameterize the temporal LAI and GPP variability and its application to describe the spatial patterns of LAI, GPP and phenological properties of deciduous broadleaf forests across eastern United States. Adequate parameterization of land surface and vegetation properties is a very important scientific task for modern biogeochemistry. New algorithms can be very useful to solve different applied problems related to adequate description of the land surface - atmosphere interaction in different spatial and temporal scales.

In the paper authors showed new modeling results and their comparison with data obtained using previously developed approaches. Obtained new results however didn't show any significant accuracy improvement in GPP estimations. The difference between measured (derived from measured NEE) and simulated GPP (fig. 3) is still very high.

Other point for discussion is model assumptions used in the study. In particular authors assumed a linear relationship between the steady-state LAI and the corresponding GPP. However in reality the GPP is non-linearly depended on LAI (not only on total LAI but even on LAI of green biomass) mainly due to non-linear PAR (photosynthetically active radiation) interception within a plant canopy. Such effects are especially pronounced in dense plant canopies with a high LAI. GPP rate is linked with forest and tree architecture. The leaf photosynthesis properties are also varied among different vegetation types. The assimilation rate is depended on biophysical properties of individual plants, water availability, nutrient supply, etc. So, the correctness of made assumption in the study is not obvious and it needs additional discussion.

Authors pointed out in result chapter about a gut agreement between leaf phenology derived by new method and MODIS data. It is true. But it is not clear from the paper the reasons for available differences between tower observed time of foliage expansion (indicated in shape of black GPP curve) and corresponding time predicted by developed model (fig 1 a-b, page 11)? The model actually predicts earlier leaf onset in spring than in situ observation (GPP data).

In the first half of introduction authors used many well known statements such as e.g. "energy and mass exchange in a plant canopy can be modeled as a function of environmental conditions (e.g., sunlight, soil moisture, temperature, and humidity) and vegetation LAI "and refereed them to most recent own publications only, and not to

available synthesis studies conducted during the last several decades and focused on the same problem.

I find that the discussion chapter is too short. It should be extended. All obtained results have to be discussed in more details.

The sentence in page 4 is not clear "leaf dynamics takes days or even months in response to climate variation". I guess authors mean weather not climate variations. Time scale for climate variation is much larger.

I'm not agreed also that the term potential evapotranspiration assumes the fixed LAI (page 3) for any hypothetic canopy. Fixed LAI can be obviously used for calculation of "reference evapotranspiration" but not potential evapotranspiration. Potential evapotranspiration rate can be estimated for plant canopy with different LAI values.

---

## Referee Comment (RC2) · Anonymous Referee #2 · 4 Oct 2018

A time-stepping scheme to simulate leaf area index, phenology, and gross primary production across deciduous broadleaf forests in eastern United States

Qinchuan Xin, Yongjiu Dai, and Xiaoping Liu

Authors present a scheme which can determine LAI for implementation in land surface models and illustrate its usefulness using the light use efficiency based production model. The paper is reasonably written but the scheme proposed is not as novel or well justified as the authors claim. In my opinion the manuscript needs a major rewrite to bring out the usefulness of authors' scheme while keeping in mind the factors mentioned below.

My major comment is that it is not justified why equilibrium LAI should be a linear function of GPP. While it is certainly easy to do so and it is attractive from a modelling perspective – can the authors compile some empirical observations to justify this assumption. Second, much stress is laid on the new scheme which can determine LAI as the model runs forward in time. This is likely because authors' previous model did not do so. However, almost all land surface models which are implemented in climate models do so already. There is nothing unique about finding d(LAI)/dt on the fly as the model moves forward in time. As such then stressing "time stepping" in manuscript's title seems inappropriate. Third, the current land surface models used in climate models have phenology schemes which are already more complicated than what the authors' have proposed in this manuscript so the tone established in the Introductory section is also not entirely correct. What authors have proposed is a very simple and easy to understand phenology scheme. Simplicity is always appreciated as long as authors are aware of the limitations of their approach and these limitations are properly identified and documented. Finally, I am unclear about how the approach used by the authors can be applied in a modelling world where a model moves forward through time driven with meteorological data. For example, on Page 6 (line 26) authors say "Given the modelled LAI time series, both vegetation phenology and GPP can be easily retrieved". The use of the word "retrieved" is confusing. In a model, GPP depends on current LAI and the current time step's GPP is used to determine LAI for the next time step. It is unclear how this can be achieved in authors' framework.

I have several other minor comments and these are marked on an annotated version of the manuscript which I attach as a PDF file.

Please also note the supplement to this comment:
https://www.biogeosciences-discuss.net/bg-2018-383/bg-2018-383-RC2- supplement.pdf

**Supplement:**

[revised manuscript text omitted]

*Good intro but I hope more detail will follow*

[Figure]

*→ if you r referring to regional and global climate simulations then these quantities are available.*

radiative transfer, leaf stomatal conductance, leaf transpiration, leaf photosynthesis, and soil evaporation, is computationally intensive and requires various climate input data that are often not readily available for regional to global simulations.

Aiming to solve the above-mentioned problems, the objectives of the study are to: 1) develop a time stepping scheme to
simulate leaf dynamics and vegetation productivity, and 2) simplify the GPD model to allow for long-term applications at a large scale. Given that the phenology modeling in deciduous broadleaf forest, a biome that have distinct seasonal growing cycles, still has large uncertainties (Melaas et al., 2016), this study choose to simulate the deciduous broadleaf forests across the eastern United States such that the developed method if successful could provide a potential basis for future applications to other biomes.

**2 Methods and materials**

**2.1 Modeling steady-state leaf area index**

One difficulty in vegetation phenology modeling is that the time scale associated with leaf allocation far exceeds that of many other vegetation processes. Unlike leaf photosynthesis that approaches equilibrium within a minute and stomatal functioning that reaches the steady state in minutes (Sellers et al., 1996a), leaf dynamics takes days or even months in response to climate
variation (Zeng et al., 2013). (Xin et al., 2018) first put forward the concept of the steady-state leaf area index, i.e., canopy LAI when time approaches infinity while the environmental conditions remain unchanging. An alternative biological explanation to the steady-state LAI is the maximum canopy LAI that an environment can sustain infinitely by its own photosynthetic activities. Supposing that the carrying capacity of canopy LAI is proportional to total canopy photosynthetic rate under a given environment, the steady-state LAI can be modeled as follows:

$$LAI_s = mGPP_s \tag{1}$$

where $LAI_s$ denotes the steady-state leaf area index; m denotes the constant ratio of steady state leaf area index to environmental capacity; and $GPP_s$ denotes the steady-state gross primary production.

The above equation, despite having a simple form, provides a critical function that complements the canopy photosynthesis model. The only parameter m is dependent on plant functional type and can be quantified from field measurements as the
average ratio of LAI to GPP at canopy closure (i.e., the time when both canopy LAI and GPP reach equilibrium). Studies have developed various canopy photosynthesis models, such as the light use efficiency models and the process-based models. Our previous studies implemented a sophisticated canopy model that assembles the sub-models of canopy radiative transfer, leaf stomatal conductance, leaf transpiration, soil evaporation, and leaf photosynthesis. Although the method has been successfully applied to different biomes, the model structure is complicated for studies at the regional to global scales. To simulate canopy
photosynthesis, this study implements the MOD17 algorithm, a big-leaf light use efficiency model that is used to provide

*uses*

*→ pls give references*

routine satellite products (Running et al., 2004). The use of the MOD17 algorithm could greatly simplify the modeling processes and reduce the required climate variables, thereby allowing for broad applications. A brief description on the MOD17 algorithm is provided here where details can be found from the user guide of the MODIS GPP product (Running and Zhao, 2015).

Based on the MOD17 algorithm, vegetation GPP can be modeled as follows:

$$GPP_s = PAR \times FPAR \times \varepsilon_{max} \times f(T) \times f(VPD) \tag{2}$$

where $GPP_s$ denotes the steady-state gross primary production; PAR denotes photosynthetically active radiation; FPAR denotes the fraction of photosynthetically active radiation; $\varepsilon_{max}$ denotes maximum light use efficiency; and $f(T)$ and $f(VPD)$ denote the scalar functions that account for the limitation of temperature and vapor pressure deficit, respectively, on canopy photosynthesis.

The fraction of photosynthetically active radiation can be modeled as follows (Turner et al., 2006):

$$FPAR = 1 - \exp(-kLAI_s) \tag{3}$$

where k denotes the canopy light extinction coefficient and $LAI_s$ denotes the steady-state leaf area index.

The environmental scalars can be modeled as follows:

[Figure]

Does this mean photosynthesis is not limited by Tmax? What if it's 50°C? Will trees still photosynthesize?

$$f(T) = max\left(min\left(\frac{TMIN - TMIN_{min}}{TMIN_{max} - TMIN_{min}}, 1\right), 0\right) \tag{4}$$

$$f(VPD) = max\left(min\left(1 - \frac{VPD - VPD_{min}}{VPD_{max} - VPD_{min}}, 1\right), 0\right) \tag{5}$$

where TMIN denotes daily minimum air temperature; $TMIN_{min}$ and $TMIN_{max}$ denote the lower and upper thresholds of daily minimum air temperature for vegetation photosynthetic activities, respectively; VPD denotes daily vapor pressure deficit; and $VPD_{min}$ and $VPD_{max}$ denote the lower and upper thresholds of daily vapor pressure deficit for vegetation photosynthetic activities, respectively.

Given the environmental conditions, Equations 1 and 2 together form simultaneous equations. Because $LAI_s$ increases as a linear function of $GPP_s$ in Equation 1 and $GPP_s$ increases as a logarithmic function of $LAI_s$ in Equation 2, the simultaneous equations have one and only one nonzero solution of $LAI_s$. The nonzero solution can be obtained by implementing a numerical approach that starts with a given initial value of $LAI_s$ and then solves the equations iteratively until converging.

*reword.
unclear sentence.*

**2.2 Modeling leaf area index, phenology, and gross primary production**

Because the physiological processes that plants allocate leaf biomass do not respond instantaneously to climate variation, there is a need to simulate vegetation LAI as lagging behind the steady state. One method to account for the time lagging effect is to apply the simple moving average method to buffer abrupt changes from individual events in the time series. Our previous study applied the simple moving average method to model LAI as the unweighted mean of the previous $LAI_s$ as follows (Xin et al., 2018):

$$LAI = \frac{1}{n_{day}} \sum_{i=0}^{n_{day}-1} LAI_s \qquad (6)$$

where LAI denotes leaf area index at the n day; $n_{day}$ denotes the number of days; i denotes an index starting from 0 to $n_{day}$ − 1; and $LAI_s$ denotes the steady state leaf area index.

The simple moving average method, while showing useful in vegetation phenology modeling, is suitable for retrospective analysis rather than prediction, and importantly, it does not match with most land surface models that operate at incremental time steps. Analogous to the method used to simulate leaf stomatal conductance in response to environmental variation, this study proposes a time stepping scheme to simulate LAI realistically as lagging behind the steady state by a simple restricted growth model (Sellers et al., 1996a) as follows:

$$\frac{dLAI}{dt} = k_l(LAI_s - LAI) \qquad (7)$$

where t denotes the time; $k_l$ denotes a time constant that reflects the responses of plant leaf allocation to climate variation; and LAI and $LAI_s$ denote the leaf area index and the steady state leaf area index, respectively.

In the time stepping scheme, vegetation LAI does not change much during winter or summer as the current LAI is close to $LAI_s$, whereas vegetation LAI increases (or decreases) during spring (or autumn) as the current LAI is less (or greater) than

$LAI_s$. For example, when the environment turns favorable for plant growth in spring, $LAI_s$ exceeds LAI and dLAI/dt is positive such that the modeled canopy LAI increases. Note that the method developed here essentially uses the canopy photosynthetic capacity (i.e., the steady-state gross primary production) instead of air temperature as a synthesized indicator to track the suitability of the environment to plant growth in time series, and therefore, the developed method is referred to as the Simplied Growing Production-Day (SGPD) model following our previous studies (Xin et al., 2018).

*How can GPP be retrived if GPP & LAI depend on each other?   Also do you mean 'modelled' instead of 'retrived'?*

Given the modeled LAI time series, both vegetation phenology and canopy GPP can be easily retrieved. Various approaches have already been developed to derive the timing of key phenophases such as spring onset and autumn senescence from seasonal LAI trajectories. This study models the phenological transition dates using a simple method that derives the first

spring and last autumn dates at which LAI reaches 20%, 50%, and 80% of the seasonal amplitudes (Richardson et al., 2012).

The selected relative amplitudes (20%, 50%, and 80%) are correspondent to different plant growth stages over a growing season. Given temperature and vapor pressure deficit, the canopy GPP is simply modeled by substituting the modeled LAI

time series back into the MOD17 algorithm. *I'm still unclear how the whole thing works. E.g. on first day of spring when LAI is still zero what do you use to calculate GPP?*

**2.3 Comparative studies using Growing Season Index**

The Growing Season Index (GSI), a widely used method in vegetation phenology modeling, allows for modeling seasonal LAI

time series rather than individual phenophases and is implemented to make direct comparisons with the SGPD model (Jolly et *Wrong place to put Jolly et al. reference.*

al., 2005). The GSI model performs comparably to or even outperforms other terrestrial biosphere models on predicting the timing of key phenophases for deciduous broadleaf forests (Melaas et al., 2013). *SGPD is your model.*

The instantaneous GSI is first derived based on the work of (Jolly et al., 2005) as follows:

$$iGSI = iTMIN \times iVPD \times iPhoto \tag{8}$$

where iGSI denotes instantaneous growing season index; and iTMIN, iVPD, and iPhoto denote the instantaneous scalar functions that account for the constraints of daily minimum air temperature, vapor pressure deficit, and photoperiod, respectively, on vegetation growth. The scalar functions for iTMIN, iVPD, and iPhoto have the mathematic forms similar to

Equations 4 and 5 and are derived the same as defined in (Jolly et al., 2005). *Please show the functions here since you will be comparing your approach to*

LAI can be modeled as the simple moving average of the instantaneous GSI scaled using maximum LAI as follows: *this .*

$$GSI = \frac{1}{n_{day}} \sum_{i=0}^{n_{day}-1} iGSI \tag{9}$$

$$LAI = GSI \times LAI_{max} \tag{10}$$

where GSI denotes growing season index at the n day; $n_{day}$ denotes the number of days; i denotes an index starting from 0 to the previous one day; iGSI denotes the instantaneous growing season index; and $LAI_{max}$ denotes the maximum leaf area index at canopy closure.

It is noteworthy that the instantaneous GSI uses the product of the scalars of minimum temperature, vapor pressure deficit, and photoperiod as an indicator to track the potential canopy photosynthetic capacities on the daily basis. Both the GSI model and the SGPD model, despite having different forms, share the same modeling idea. To understand the differences between the simple moving average method and the time stepping method, the GSI model is also implemented with the simple restricted growth model as follows:

[Figure]

$$LAI_s = iGSI \times LAI_{max} \tag{11}$$

$$\frac{dLAI}{dt} = k_1(LAI_s - LAI) \tag{12}$$

[revised manuscript text omitted]

*Rewrite this to say that you used two satellite LAI products. Then say that the MODIS one was used for site level studies as well.*

[Figure]

**2.5 Model parameterization and comparison**

As the MOD17 algorithm is a well-parameterized model, this study applies the model parameters *from* literature directly. Following the user guide of the MODIS GPP product (Running and Zhao, 2015), key parameters in the MOD17 algorithm are set as $\varepsilon_{max}$ = 1.165 gC/MJ , $TMIN_{min}$ = −6.0 °C, $TMIN_{max}$ = 9.94 °C, $VPD_{min}$ = 0.65 kPa, and $VPD_{max}$ = 1.65 kPa. The light extinction coefficient of the canopy is 0.5. The parameter that defines the ratio of leaf area index to environmental capacity is set as m = 0.58 $m^2$(leaf area)/gC/day as quantified using the average ratio of LAI to GPP at canopy closure using the flux tower data. The canopy maximum LAI is set as 5.80 based on the maximum 95th percentile of satellite-derived LAI across sites and years (Xin et al., 2018). The parameter $n_{day}$ in the simple moving average method and the parameter $k_t$ in the time stepping method control the response of plant leaf allocation to environmental variation. The parameter $n_{day}$ is set as 21 days and the parameter $k_t$ is calibrated as 0.080 $day^{-1}$.

*It should have been mentioned earlier that your approach gives you 4 different ways to calculate LAI.*

[revised manuscript text omitted]

*Wouldn't it be better to show annual maximum LAI, or LAI averaged over the growing season*

**3.2 Regional-scale modeling**

Figure 4 shows the spatial extents of the 10-year (2001-2010) mean LAI and associated errors as derived from remote sensing data and model simulations. The SGPD-TS method could well capture the spatial pattern of the satellite-derived LAI, including the decreasing gradients from south to north and the decreases in mountain areas (Figure 4a and 4b). The 10-year mean LAI derived from the GSI-TS method (Figure 4c) also show a decreasing trend from south to north but the modeled LAI is much larger than the MODIS LAI. Because the GSI-TS method defines the maximum leaf area index for the growing season, the

overestimation on the modeled 10-year mean LAI is primarily due to model overestimates in the spring and autumn phenological transitions. As compared with the MODIS observations, RMSE and MBE obtained by the SGPD-TS method are much smaller than and distribute more evenly than those obtained by the GSI-TS method. RMSE for the GSI-TS LAI exhibit a decreasing north-south gradient, implying that the model accuracies are lower in southern areas lower than in northern areas.

MBE for the GSI-TS model are greater than 0.5 m²/m² for most areas. When comparing SGPD-TS LAI with MODIS LAI, RMSE are less than 0.5 m²/m² and MBE are minor across the study region. The amplitudes of the error metrics in the regional-scale studies are consistent with those in the site-scale studies. Note that some studies applied the multi-year mean LAI as derived from the remote sensing data to simulate the land surface processes, the results obtained here indicate that the SGPD-TS method can be used alternatively to provide multi-year mean LAI time series via climate variables for land surface studies.

*I'm still a bit confused. If LAI & GPP depend on each other how can this be done.*

[revised manuscript text omitted]

_What about $CO_2$? GPP also depends on $CO_2$._

_( How will one find $m$ for all PFTs ?_

method could fit seamlessly into the land surface models that operate at incremental time steps such as the Community Land Model and the Common Land Model (Dai et al., 2003). Because the state-of-the-art land surface models all include the canopy photosynthesis sub-model, the developed method can then be easily embedded into these land surface models as an alternative phenology model. Compared to the simple light use efficiency model like the MOD17 algorithm, implementation of the developed time-stepping scheme in the land surface models relies on supercomputing for global applications. To better understand the performance of the develop method, one study is now undertaken to implement the developed method with the Common Land Model for simulating multi-decadal LAI and GPP for global biomes via only climate variables.

_forced      by_

Applying the developed method to other biomes and other regions still has issues to be solved appropriately. The time stepping method uses the parameter $k_1$ to account for the time lags of leaf allocation in response to environmental changes. For the deciduous broadleaf forests, a biome with strong seasonality, the developed scheme achieved reasonable results with appropriate parameterization. Short vegetation like grasslands tends to respond much quickly to abrupt environment changes like precipitation and tropical ecosystems have strong resilience to short-term environmental variation (Shen et al., 2011;Levine et al., 2016). These understandings from the observational studies imply that biomes have varied response speeds to the environment and proper model calibration and assessment are required for the developed method. Using the observation data from remote sensing alone is inadequate for model development as satellite-derived LAI could have large uncertainties for some specific biomes other than deciduous broadleaf forests. Fortunately, global flux tower network and regional phenology observation networks are now established and offer abundant data for comprehensive model assessment.

_how does a linear relationship between LAI & GPP support this_

**5 Conclusions       _provide_**

Terrestrial biosphere models is a basic tool for understanding the interactions between the land surface and the atmosphere. To provide a complete solution to the simulation of plant leaf dynamics and canopy photosynthesis, this study establishes a linear relationship between the steady state leaf area index and the corresponding canopy photosynthetic capacity based on the idea that plants optimize leaf biomass allocation according to the environmental carrying capacity for maximizing photosynthetic carbon gain. The proposed leaf allocation function complements the canopy photosynthesis model of the

MOD17 algorithm to form simultaneous equations that can be solved iteratively. To account for the time lagging of plant leaf allocation in response to climate variation, a time stepping scheme based on a simple restricted growth model is applied to the solved steady state leaf area index to obtain time series of leaf area index. The developed method could perform reasonably well on simulating leaf area index, phenology, and gross primary production for deciduous broadleaf forests across eastern United States over years as found in both the site-scale and regional-scale modeling studies. Compared to the simple moving average method, the time stepping scheme developed here is consistent with and can be easily embedded into the state-of-the-art land surface models that typically operate at incremental time steps. The developed method allows for simulating leaf area

_This wasn't shown explicitly_

---

## Author Comment (AC1) · 13 Nov 2018

Dear reviewer,

Thank you for taking time to review our manuscript.
We studied your comments and revised our draft accordingly.
We also went through our manuscript and made corrections on the texts.
Changes to the texts were marked in the manuscript and were presented as follows.
Hope that our revised draft will meet with your approval.

Best wishes,
Qinchuan Xin

**Anonymous Referee # 1**

The study provided by Q. Xin et al "A time-stepping scheme to simulate leaf area index, phenology, and gross primary production across deciduous broadleaf forests in eastern United States" is mainly focused on development of a new modeling algorithm to parameterize the temporal LAI and GPP variability and its application to describe the spatial patterns of LAI, GPP and phenological properties of deciduous broadleaf forests across eastern United States. Adequate parameterization of land surface and vegetation properties is a very important scientific task for modern biogeochemistry. New algorithms can be very useful to solve different applied problems related to adequate description of the land surface - atmosphere interaction in different spatial and temporal scales.

In the paper authors showed new modeling results and their comparison with data obtained using previously developed approaches. Obtained new results however didn't show any significant accuracy improvement in GPP estimations. The difference between measured (derived from measured NEE) and simulated GPP (fig. 3) is still very high.

**Reply:** Thank you for your comments. This study does not try to improve the GPP simulation but tries to develop a method that can use climate variables to simulate both LAI and GPP. For land surface models that predict GPP values, they require either satellite-derived LAI data or an independent phenology sub-model. The main idea for this study is to improve the phenology modeling by providing time series of LAI simulated using climate variables. Because here we implement the MOD17 algorithm instead of the sophisticated process-based model for the purpose of model simplicity, we have no expectation that the GPP simulated based on model-simulated LAI could be more accurate than the GPP simulated based on satellite-observed LAI.

To address your concern, we add sentences to main texts in the discussion part as follow:
Land surface models that predict vegetation GPP require either satellite-derived LAI input data or the phenology sub-model. The main idea for this study is to improve the phenology modeling by providing time series of LAI simulated using climate variables, and hence enables to simulate GPP forced only by climate variables. Because we implement the MOD17 algorithm instead of the sophisticated

process-based model for the purpose of simplicity, one should not expect that GPP simulated based on the model-simulated LAI could be more accurate than GPP simulated based on the satellite-derived LAI.

Other point for discussion is model assumptions used in the study. In particular authors assumed a linear relationship between the steady-state LAI and the corresponding GPP. However in reality the GPP is non-linearly depended on LAI (not only on total LAI but even on LAI of green biomass) mainly due to non-linear PAR (photosynthetically active radiation) interception within a plant canopy. Such effects are especially pronounced in dense plant canopies with a high LAI. GPP rate is linked with forest and tree architecture. The leaf photosynthesis properties are also varied among different vegetation types. The assimilation rate is depended on biophysical properties of individual plants, water availability, nutrient supply, etc. So, the correctness of made assumption in the study is not obvious and it needs additional discussion.

**Reply:** We agree with your comments that additional discussion is needed. We conducted an experiment and added paragraphs in the discussion to show the relationship between leaf area index and other variables on the monthly basis. Note that our modeling approach does not try to model GPP based on LAI, but tries to model LAI as a function of GPP. This provides another key function to close the equation systems such that two unknown variables (i.e., LAI and GPP at the steady state) in two independent equations can then be solved numerically. The added paragraphs in the discussion are as follows:

Figure 8 further illustrate the relationship between mean LAI and different variables on a monthly basis. All data were averaged to the monthly time scale such that canopy LAI can be considered as nearly the steady state. On the monthly basis, mean LAI has a strong near-linear relationship with mean GPP ($R^2$=0.888) and the slope for the regression without intercept is 0.580, the same as we used in the model simulation. On the monthly basis, mean LAI is strongly correlated with mean temperature ($R^2$=0.799), indicating that temperature is the dominate factor that determines vegetation phenology. Factors like vapor pressure deficit and photoperiod also have positive relationships with mean LAI on the monthly basis. Figure 8 suggests that the processes of leaf phenology and photosynthetic phenology for deciduous broadleaf forest are closely related. Our modeling approach that links canopy GPP with LAI reflects the empirical positive relationship found in Figure 8a.

[Figure]

Figure 8: Scatter plots are shown for the relationship a) between mean leaf area index and mean gross primary production, b) between mean leaf area index and mean temperature, c) between mean leaf area index and mean vapor pressure deficit, and d) mean leaf area index and mean photoperiod on a monthly basis. All available site-year flux tower data were included in the analysis. All data were averaged to the monthly time scale for analysis. The dashed lines denote the regression lines. Figure 8a uses the regression without intercept.

Authors pointed out in result chapter about a gut agreement between leaf phenology derived by new method and MODIS data. It is true. But it is not clear from the paper the reasons for available differences between tower observed time of foliage expansion (indicated in shape of black GPP curve) and corresponding time predicted by developed model (fig 1 a-b, page 11)? The model actually predicts earlier leaf onset in spring than in situ observation (GPP data).

**Reply:** Thank you for pointing out that the model actually predicts earlier leaf onset in spring than in situ observation in terms of the GPP data. We suspect that the flux tower GPP data as shown in the figure suffer from instrument malfunction, because the GPP data increased sharply and unreasonably within very short time in the spring time. To avoid the confusion, we update the figure using data from US-UMB in 2004.

The updated figure looks as follows:

[Figure]

Figure 1: The modeled and measured daily time series of a) leaf area index and b) gross primary production are shown for the flux tower site of US-UMB in 2004. The reference LAI time series in Figure 1a are derived from the MODIS data and the reference GPP time series in Figure 1b are obtained from the flux tower measurements.

In the first half of introduction authors used many well known statements such as e.g. "energy and mass exchange in a plant canopy can be modeled as a function of environmental conditions (e.g., sunlight, soil moisture, temperature, and humidity) and vegetation LAI "and refereed them to most recent own publications only, and not to available synthesis studies conducted during the last several decades and focused on the same problem.

**Reply:** Thank you for your comments. We removed the self-cited references and added some recent studies to the texts as follows:

Hufkens, K., Basler, D., Milliman, T., Melaas, E. K., and Richardson, A. D.: An integrated phenology modelling framework in R, Methods in Ecology and Evolution, 9, 1276-1285, 2018.

Li, W., Guo, Q., Tao, S., and Su, Y.: VBRT: A novel voxel-based radiative transfer model for heterogeneous three-dimensional forest scenes, Remote Sensing of Environment, 206, 318-335, 2018.

Liu, Q., Fu, Y. H., Liu, Y., Janssens, I. A., and Piao, S.: Simulating the onset of spring vegetation growth across the Northern Hemisphere, Global change biology, 24, 1342-1356, 2018.

Yuan, H., Dickinson, R. E., Dai, Y., Shaikh, M. J., Zhou, L., Shangguan, W., and Ji, D.: A 3D Canopy Radiative Transfer Model for Global Climate Modeling: Description, Validation, and Application, Journal of Climate, 27, 1168-1192, 2013.

I find that the discussion chapter is too short. It should be extended. All obtained results have to be discussed in more details.

**Reply:** Thank you for your comments. We agree that the discussion chapter should be extended. Based on other comments, we further discussed the relationship between leaf area index and other climate variables and we discussed the model limitations. Please find our responses mentioned above and in the manuscript for details to changes in the discussion chapter.

The sentence in page 4 is not clear "leaf dynamics takes days or even months in response to climate variation". I guess authors mean weather not climate variations. Time scale for climate variation is much larger.

**Reply:** We agree that using the word "weather" is precise. We revised the sentence as "Unlike leaf photosynthesis that approaches equilibrium within a minute and stomatal functioning that reaches the steady state in minutes (Sellers et al., 1996a), leaf dynamics takes days or even months in response to weather variation (Zeng et al., 2013)."

I'm not agreed also that the term potential evapotranspiration assumes the fixed LAI (page 3) for any hypothetic canopy. Fixed LAI can be obviously used for calculation of "reference evapotranspiration" but not potential evapotranspiration. Potential evapotranspiration rate can be estimated for plant canopy with different LAI values.

**Reply:** Thank you for your comments. We agree that using "reference evapotranspiration" is more accurate than using "potential evapotranspiration". Note that some researchers still use "potential evapotranspiration" in their publications as the term "potential evapotranspiration" is easy to understand. Based on your suggestions, we revised the sentence as "
[revised manuscript text omitted]

---

## Author Comment (AC2) · 13 Nov 2018

Dear reviewer,

Thank you for taking time to review our manuscript.
We studied your comments and revised our draft accordingly.
We also went through our manuscript and made corrections on the texts.
Changes to the texts were marked in the manuscript and were presented as follows.
Hope that our revised draft will meet with your approval.

Best wishes,
Qinchuan Xin

**Anonymous Referee # 2**

Authors present a scheme which can determine LAI for implementation in land surface models and illustrate its usefulness using the light use efficiency based production model. The paper is reasonably written but the scheme proposed is not as novel or well justified as the authors claim. In my opinion the manuscript needs a major rewrite to bring out the usefulness of authors' scheme while keeping in mind the factors mentioned below.

My major comment is that it is not justified why equilibrium LAI should be a linear function of GPP. While it is certainly easy to do so and it is attractive from a modelling perspective – can the authors compile some empirical observations to justify this assumption.

**Reply:** Thank you for your comments. We agree that there is a need to provide some empirical observation to justify the assumption that equilibrium LAI should be a linear function of GPP. We conducted a study on the monthly basis and added paragraphs in the discussion chapter.

Figure 8 further illustrate the relationship between mean LAI and different variables on a monthly basis. All data were averaged to the monthly time scale such that canopy LAI can be considered as nearly the steady state. On the monthly basis, mean LAI has a strong near-linear relationship with mean GPP ($R^2=0.888$) and the slope for the regression without intercept is 0.580, the same as we used in the model simulation. On the monthly basis, mean LAI is strongly correlated with mean temperature ($R^2=0.799$), indicating that temperature is the dominate factor that determines vegetation phenology. Factors like vapor pressure deficit and photoperiod also have positive relationships with mean LAI on the monthly basis. Figure 8 suggests that the processes of leaf phenology and photosynthetic phenology for deciduous broadleaf forest are closely related. Our modeling approach that links canopy GPP with LAI reflects the empirical positive relationship found in Figure 8a.

[Figure]

Figure 8: Scatter plots are shown for the relationship a) between mean leaf area index and mean gross primary production, b) between mean leaf area index and mean temperature, c) between mean leaf area index and mean vapor pressure deficit, and d) mean leaf area index and mean photoperiod on a monthly basis. All available site-year flux tower data were included in the analysis. All data were averaged to the monthly time scale for analysis. The dashed lines denote the regression lines. Figure 8a uses the regression without intercept.

Second, much stress is laid on the new scheme which can determine LAI as the model runs forward in time. This is likely because authors' previous model did not do so. However, almost all land surface models which are implemented in climate models do so already. There is nothing unique about finding d(LAI)/dt on the fly as the model moves forward in time. As such then stressing "time stepping" in manuscript's title seems inappropriate.

**Reply:** Thank you for your valuable advice. We agree that most land surface models run forward in time. The idea here is to provide a simple solution to do so. To address your concerns, we revised the title as "A simple time-stepping scheme to simulate leaf area index, phenology, and gross primary production across deciduous broadleaf forests in eastern United States".

Third, the current land surface models used in climate models have phenology schemes which are already more complicated than what the authors' have proposed in this manuscript so the tone established in the Introductory section is also not entirely correct. What authors have proposed is a very simple and easy to understand phenology scheme. Simplicity is always appreciated as long as authors are aware of the limitations of their approach and these limitations are properly identified and documented.

**Reply:** Thank you for your valuable advice. We have revised the introduction part by removing the sentences that we consider inappropriate. We also added sentences to the discussion part and identified the model limitations. The added sentences are as follows:

The MOD17 algorithm only assumes the monotonic relationship between air temperature and photosynthesis and between vapor pressure deficit and photosynthesis. It also does not account for the impacts of CO2 on photosynthesis. The use of the MOD17 algorithm in this study thus has limitations in the model structure.

Land surface models that predict vegetation GPP require either satellite-derived LAI input data or the phenology sub-model. The main idea for this study is to improve the phenology modeling by providing time series of LAI simulated using climate variables, and hence enables to simulate GPP forced only by climate variables. Because we implement the MOD17 algorithm instead of the sophisticated process-based model for the purpose of simplicity, one should not expect that GPP simulated based on the model-simulated LAI could be more accurate than GPP simulated based on the satellite-derived LAI.

Finally, I am unclear about how the approach used by the authors can be applied in a modelling world where a model moves forward through time driven with meteorological data. For example, on Page 6 (line 26) authors say "Given the modelled LAI time series, both vegetation phenology and GPP can be easily retrieved". The use of the word "retrieved" is confusing. In a model, GPP depends on current LAI and the current time step's GPP is used to determine LAI for the next time step. It is unclear how this can be achieved in authors' framework.

Reply: In our approach, we first solve the current steady-state leaf area index using the current meteorological data. Note that the steady-state leaf area index is not the actual leaf area index. Both the actual leaf area index and the steady-state leaf area index are used to calculate the actual leaf area index at the next time step in our framework. Once the actual leaf area index at the next step is known, GPP at the next time step can be modeled using the meteorological data at the next time step based on the MOD17 algorithm.

At day zero, the very first beginning of a time series, the solved current steady-state leaf area index can be used as surrogate to the current actual leaf area index, and such that both are used to derive the actual leaf area index at the next time step. The error is negligible because the solved current steady-state leaf area index is often very close to zero for deciduous broadleaf forest during wintertime as the air temperature is low.

Another issue that you frequently questioned is how to obtain values for LAI and GPP given that LAI and GPP are dependent on each other. We have two unknown variables (i.e., LAI and GPP at the steady state) and two different generalized equations, and this is the situation that we called a closed system of equations. When the two equations are simple, one may derive an analytic solution (or the closed form solution). But here is not the case, because the dependence of GPP on LAI is non-linear and complicated. The numerical approach is to give a guess value initially and then iterates to obtain an approximate solution when the solution is converging. Note that this method is similar to what the Community Land Model 4.5 uses to solve stomatal resistance and leaf photosynthesis. When the stomatal resistance ($r_s$), the $CO_2$ partial pressure at the leaf surface ($c_s$), the internal leaf $CO_2$ partial pressure ($c_i$) and the leaf net photosynthesis ($A_n$) are dependent on each other, the Community Land Model 4.5 applies the numerical approach to solve their values iteratively until the internal leaf $CO_2$ partial pressure ($c_i$) converges.

We agree that the word "retrieved" is confusing and we now use "modeled" instead.

I have several other minor comments and these are marked on an annotated version of the manuscript which I attach as a PDF file.
**Reply:** Please find our detailed responses to your comments as follows.

P1 L9: suggested changes to texts
**Reply:** Based on your comments, we deleted "Robust" and "both time series of". It now reads as "Terrestrial biosphere models that simulate leaf dynamics and canopy photosynthesis are required to understand the vegetation-climate interactions."

P1 L11: suggested changes to texts
**Reply:** As you suggested, we replaced "simultaneously via only" with "when forced with".

P1 L12-14: The sentence "plants allocate leaf biomass till an environment could sustain to maximize photosynthetic reproduction. The method establishes a linear function between the steady-state LAI and the corresponding GPP, which is used to track the suitability of environmental conditions for plant photosynthesis, and applies the MOD17 algorithm to form" is unclear.
**Reply:** We revised the sentences. It now reads as "The method establishes a linear function between the steady-state LAI and the corresponding GPP, which is used to track the suitability of environmental conditions for plant photosynthesis. The method applies the established function and the MOD17 algorithm to form simultaneous equations together, which can be solved numerically."

P1 L15-16: "leaf allocation to environment variation" is unclear.
**Reply:** we revised the sentence as "time-lagged responses of plant growth to environmental conditions"

P1 L26: "momentum flows on the land surface" is unclear.

**Reply:** Based on your suggestion, we deleted the related part. It now reads as "The canopy structures and characteristics govern solar radiation interception and absorption (Ni-Meister et al., 2010; Yuan et al., 2013)."

P1 L27: suggested changes to texts

**Reply:** As you suggested, we deleted "Individual". The sentence now reads as "Plants control water transpiration and photosynthetic carbon fixation through processes from transient changes in leaf stomatal conductance to seasonal variation in foliage dynamics (Eagleson, 2005)."

P2 L1: suggested changes to texts

**Reply:** Based on your comments, we replaced "model" and "integrates" with "models" and "integrate", respectively. We also replaced "knowledges of the Earth science is an essential tool" with "knowledge of Earth sciences allow".

It now reads as "Numerical terrestrial biosphere models that integrate multidisciplinary knowledge of Earth sciences allow to understand and predict the interactions between terrestrial ecosystems and the climate under a changing global environment."

P2 L5: To match in what way? Representation of what?

**Reply:** To make it clear, we revised the sentence as follows: Developments on the terrestrial biosphere models essentially seek accurate solution to the simulation of energy and material exchanging fluxes between ecosystems and the atmosphere.

P2 L10: suggested changes to texts

**Reply:** We replaced "As the vigorous" with "The" and added "and". The sentence reads as "The development of satellite remote sensing technology offers large-scale observations for vegetation monitoring and a number of modeling approaches have been developed to quantify and simulate the land surface fluxes based on climate variables and satellite-derived LAI."

P2 L13-15: Expand acronyms.

**Reply:** Thank you for your comment. We add explanations to the acronyms. It now reads as follows: "These methods, including both the light use efficiency models (e.g., the Carnegie-Ames-Stanford Approach (CASA) model (Potter et al., 1993), the MOD17 algorithm (Running et al., 2004), the Vegetation Photosynthesis Model (VPM) (Xiao et al., 2004), the eddy covariance light use efficiency (EC-LUE) model (Yuan et al., 2010), and the two-leaf light use efficiency (TL-LUE) model (He et al., 2013)) and the process-based models (e.g., the boreal ecosystem productivity simulator (BEPS) model (Liu et al., 1997), the Breathing Earth System Simulator (BESS) model (Ryu et al., 2011), the Growing Production-Day (GPD) model (Xin, 2016), the revised Simple Biosphere (SiB2) model (Sellers et al., 1996b)), despite differing from each other on the representation of vegetation processes, have been successfully used for applications from field to global scales."

P2 L21: suggested changes to texts

**Reply:** We replaced "phenology, of which the modeling" with "phenological processes. This modeling". It now reads as "Modeling vegetation leaf dynamics via climate variables requires in-depth understanding on plant phenological processes. This modeling is still largely empirical to date and contributes considerable uncertainties to current terrestrial biosphere models (Richardson et al., 2012)."

P3 L1: What does this exactly mean? Is this amount of C allocated to leaves?
**Reply:** Your understanding is correct. To make it clear, we revised the sentence as "For example, the DeNitrification DeComposition model uses an optimal seasonal growth curve of plant LAI and then calculates environmental stresses of water and nitrogen to limit daily carbon and nitrogen allocation to plant leaves (Yu et al., 2014)."

P3 L3: suggested changes to texts
**Reply:** Based on your comments, we replaced "enriched" and "availability" with "benefitted" and "development", respectively. It now reads as "While these studies have greatly benefitted the development of the phenology models, there is still a need to improve the current phenology models."

P3 L7: No. "While most of the existing studies choose to develop the plant phenology model independently from the canopy flux model" is not entirely correct.
**Reply:** Thank you for your comments. We agree with your comments. We removed it from the texts.

P3 L11: suggested changes to texts
**Reply:** Following you suggestion, we replaced "the leaf biomass" and "variation" with "photosynthate" and "conditions", respectively. It now reads as "Given limited external resources, plants have evolved to effectively allocate photosynthate in response to environment conditions so as to maximize photosynthetic carbon gain, the fundamental bioenergy for survival (Givnish, 1986)."

P3 L15. What is the leaf distribution processes?
**Reply:** To make it clear, we revised the sentence as "In essence, synthesized analysis of both canopy photosynthesis and leaf phenology processes is needed to solve the difficulties in the development of the current terrestrial biosphere models."

P3 L20-30. Good intro but I hope more detail will follow
**Reply:** Thank you for your advice. Note that these are published results and we prefer to keep it as concise as possible. To address your concerns, we did add some more sentences to introduce the method we developed.

The related sentence reads as follows:
To allow for predicting the entire LAI time series over a growing season, (Xin et al., 2018) further improved the GPD model by proposing a linear function between LAI and GPP at the steady state. The proposed function and the sophisticated canopy GPP model (i.e., modeling GPP as a function of

LAI and climate variables) together form a closed system of equations that includes both vegetation GPP and LAI. The improved GPD model uses the numerical approach, a method that gives an initial value and then iterates to the convergence of the solution, to solve the closed system of equations and derives LAI in the steady state. The improved GPD model then applies the simple moving average method to the steady-state LAI to obtain the modeled LAI time series. The improved method circumvents the need to empirically prescribe a fixed canopy and enables modeling of LAI time series in addition to the timing of individual phenophases.

P4 L2: If you are referring to reginal and global climate simulations then these quantities are available.

**Reply:** Thank you for your advice. We deleted misleading words and it now reads as "Second, the developed GPD model that includes many subtle vegetation processes, such as canopy radiative transfer, leaf stomatal conductance, leaf transpiration, leaf photosynthesis, and soil evaporation, is computationally intensive and requires various climate input data."

P4 L27: Please give references.

**Reply:** Thank you for your advice. We added references and it now reads as "Our previous studies (Xin, 2016; Xin et al., 2018) implemented a sophisticated canopy model that assembles the sub-models of canopy radiative transfer, leaf stomatal conductance, leaf transpiration, soil evaporation, and leaf photosynthesis."

P4 L30: suggested changes to texts

**Reply:** Based on your suggestion, we replaced "is used to provide" with "uses". It now reads as "To simulate canopy photosynthesis, this study implements the MOD17 algorithm, a big-leaf light use efficiency model that uses routine satellite products (Running et al., 2004)."

P5 L16: Does this mean photosynthesis is not limited by Tmax? What if it's 50℃? Will trees still photosynthesize?

**Reply:** Thank you for your comments. Photosynthesis is indeed limited by Tmax and the photosynthesis activities will halt at high temperature. In this study, we used the MOD17 algorithm, which is used to produce the MODIS GPP product. The MOD17 algorithm only assumes the monotonic relationship between air temperature and photosynthesis and between vapor pressure deficit and photosynthesis. These are limitations in the models.

To address your concerns, we added the sentences to the discussion and acknowledged these shortcoming as follows: The MOD17 algorithm only assumes the monotonic relationship between air temperature and photosynthesis and between vapor pressure deficit and photosynthesis. It also does not account for the impacts of $CO_2$ on photosynthesis. The use of the MOD17 algorithm in this study thus has limitations in the model structure.

P6 L2: "Because the physiological processes that leaf biomass" is unclear.

**Reply:** We revised the sentence and it now reads as follows: "Because the physiological processes that plants allocate photosynthates to leaves do not respond instantaneously to climate variation, there is a need to simulate vegetation LAI as lagging behind the steady state."

P6 L26: How can GPP be retrieved if GPP & LAI depend on each other? Also do you mean "modelled" instead of "retrieved".

**Reply:** Thank you for your comments. First, we replaced "retrieved" with "modeled". Second, when GPP and LAI are dependent on each other, the solutions can be obtained using the numerical approach. To address your concerns, we revised the texts and added more explanations as follows:

Given the environmental conditions, Equations 1 and 2 together form simultaneous equations, meaning that there are two unknown variables (i.e., LAI and GPP at the steady state) and two different general equations. One may derive an analytic solution if both equations have simple forms. But because the dependence of GPP on LAI is non-linear, deriving the analytic solution is complicated and we could apply the numerical approach to obtain the solutions. Because $LAI_s$ increases as a linear function of $GPP_s$ in Equation 1 and $GPP_s$ increases as a logarithmic function of $LAI_s$ in Equation 2, the simultaneous equations have one and only one nonzero solution of $LAI_s$. To obtain the nonzero solution, the numerical approach starts with a guess value of $LAI_s$ and then then iterates to obtain the approximated solution of $LAI_s$ until converging. Note that the numerical approach is widely used in the land surface models. For example, as the stomatal resistance, the CO2 partial pressure at the leaf surface, the internal leaf CO2 partial pressure, and the leaf net photosynthesis are dependent on each other, the Community Land Model 4.5 uses the numerical approach to solve stomatal resistance and leaf photosynthesis iteratively until the internal leaf CO2 partial pressure converges.

P7 L3-4: I'm still unclear how the whole thing works. E.g. on first day of spring when LAI is still zero what do you use to calculate GPP?

**Reply:** Thank you for your comments. There are two steps in our modeling framework. The first step is to calculate the steady-state LAI and the second step is to calculate the actual LAI. We have already illustrated how to do the first step in the response above. For the second step, the actual LAI at the next time step is modeled using both the current steady-state LAI and the current actual LAI based on Equation 7. Even the current actual LAI is zero, for example, on the first day of spring, the change rates of leaf area index (dLAI/dt) is not zero because the difference between steady-state LAI and actual LAI is not zero. The model then moves forward to obtain the actual LAI at the next day.

$$\frac{dLAI}{dt} = k_l(LAI_s - LAI) \tag{1}$$

For the first day of spring, when the LAI is still zero, GPP can be obtained using LAI and climate variables based on the MOD17 model. The MOD17 model uses only daily LAI, daily minimum temperature, daily vapor pressure deficit, and daily photosynthetically active radiation as inputs to model GPP. When the LAI is zero, then the modeled GPP is zero. When LAI increases, the modeled GPP increases accordingly but is still dependent on other climate variables.

To address your concerns, we added explanations to the texts as follows:

Because the MOD17 algorithm only requires LAI, daily minimum temperature, daily vapor pressure deficit, and daily photosynthetically active radiation as model inputs, the canopy GPP is simply modeled by substituting the modeled LAI time series and the climate variables into the MOD17 algorithm. For the first day of spring when the LAI is zero, the modeled GPP is zero. As times move forward, the modeled GPP increases as LAI increases but is still dependent on other climate variables such as solar radiation, temperature and vapor pressure deficit,

P7 L6: Wrong place to put Jolly et al. reference SGPD is your model.

**Reply:** Thank you for your advice. We moved the reference to the right place. It now reads as "The Growing Season Index (GSI), a widely used method in vegetation phenology modeling (Jolly et al., 2005), allows for modeling seasonal LAI time series rather than individual phenophases and is implemented to make direct comparisons with the SGPD model."

P7 L15: Please show the functions here since you will be comparing your approach to this.

**Reply:** Thank you for your advice. We show the functions in the manuscript as follows:

The scalar functions for iTMIN, iVPD, and iPhoto have the mathematic forms similar to Equations 4 and 5 and are derived the same as defined in (Jolly et al., 2005) as follows:

$$\text{iTMIN} = \max\left(\min\left(\frac{\text{TMIN} - \text{TMIN}_{\text{min}}}{\text{TMIN}_{\text{max}} - \text{TMIN}_{\text{min}}}, 1\right), 0\right) \tag{2}$$

$$\text{iVPD} = \max\left(\min\left(1 - \frac{\text{VPD} - \text{VPD}_{\text{min}}}{\text{VPD}_{\text{max}} - \text{VPD}_{\text{min}}}, 1\right), 0\right) \tag{3}$$

$$\text{iPhoto} = \max\left(\min\left(\frac{\text{Photo} - \text{Photo}_{\text{min}}}{\text{Photo}_{\text{max}} - \text{Photo}_{\text{min}}}, 1\right), 0\right) \tag{4}$$

where TMIN denotes daily minimum temperature; $\text{TMIN}_{\text{min}}$ and $\text{TMIN}_{\text{max}}$ denote the lower and upper thresholds of daily minimum air temperature for vegetation photosynthetic activities, respectively; VPD denotes daily vapor pressure deficit; $\text{VPD}_{\text{min}}$ and $\text{VPD}_{\text{max}}$ denote the lower and upper thresholds of daily vapor pressure deficit for vegetation photosynthetic activities, respectively; Photo denotes daily photoperiod; and $\text{Photo}_{\text{max}}$ and $\text{Photo}_{\text{min}}$ denote the lower and upper thresholds of daily photoperiod for vegetation photosynthetic activities, respectively.

P8 L9: suggested changes to texts

**Reply:** We replaced "Our modeling studies are made at" with "We evaluate our approach at". It now reads as "We evaluate our approach at the site scale using both the flux tower data and remote sensing data and at the regional scale using both the climate data and remote sensing data for deciduous broadleaf forests in eastern United States."

P8 L10: suggested changes to texts

**Reply:** Following your suggestions, we replaced "In" with "For". It now reads as "For the site-scale studies, all the flux tower sites of deciduous broadleaf forests (Table 1) that are available in the AmeriFlux website (http://ameriflux.ornl.gov/) were used for analysis."

P8 L14-24: The title of this section doesn't indicate that you are talking about observations in this section.

**Reply:** Thank you for your comments. We revised the subtitle as "Study materials and pre-processing"

P9: Rewrite this to say that you used two satellite LAI products. Then say that the MODIS one was used for site level studies as well.

**Reply:** Thank you for your advice. We choose to introduce data involved in site-scale studies first and then introduce data used for regional-scale studies, because in the following section, we present the studies at different scales accordingly. For those who have interests on the details on our modeling studies, they might easily find the data involved in the corresponding studies.

P10 L1: suggested changes to texts

**Reply:** Following your suggestions, we replaced "In" with "from". It now reads as "As the MOD17 algorithm is a well-parameterized model, this study applies the model parameters from literature directly."

P10 L11-17. It should have been mentioned earlier that your approach gives you 4 different ways to calculate LAI.

**Reply:** Thank you for your suggestion. We now move the related paragraph to Section 2.4 and choose to mention the approaches provide four different ways to calculate LAI.

P10 L29: No, it seems modelled GPP is more variable than observed.

**Reply:** We removed the words that are not appropriate.

P12 L6: Existing models are already more complex.

**Reply:** Thank you for your advice. We removed the sentence.

P12 L15-16: Reword "on predicting the timing of different phenophases are listed in." and "The use of phenophases in model assessment eliminates the seasonality effects of the LAI times series." is unclear.

**Reply:** We revised the sentence. It now reads as "Table 2 lists the statistical metrics that illustrate the model performance on predicting the timing of different phenophases." We removed the sentence that is unclear.

P14 L1. Wouldn't it be better to show annual maximum LAI or LAI average as the growing season?

**Reply:** Thank you for your advice. Annual maximum LAI have large fluctuation across space and deriving the maximum LAI using our approach is largely dependent on the accuracy of the MOD17

algorithm. Deriving LAI average over the growing seasons requires to derive the spring onset and autumn senescence first. Here we only uses simple methods based on the half way of the LAI amplitude but the algorithms to derive key phenophases vary considerably in different studies. Presenting results for LAI averaging over the entire growing season does not have these problems. We have shown similar results in our previous study (Xin et al, 2018, AFM). This study uses multi-year mean values instead of one-year mean values to understand the model performance across time and space.

P15 L9: I'm still a bit confused. If LAI & GPP depend on each other, how can this be done?

**Reply:** Thank you for your comments. When GPP and LAI are dependent on each other, the solutions can be obtained using the numerical approach. To address your concerns, we revised the texts and added more explanations as follows:

Given the environmental conditions, Equations 1 and 2 together form simultaneous equations, meaning that there are two unknown variables (i.e., LAI and GPP at the steady state) and two different general equations. One may derive an analytic solution if both equations have simple forms. But because the dependence of GPP on LAI is non-linear, deriving the analytic solution is complicated and we could apply the numerical approach to obtain the solutions. Because $LAI_s$ increases as a linear function of $GPP_s$ in Equation 1 and $GPP_s$ increases as a logarithmic function of $LAI_s$ in Equation 2, the simultaneous equations have one and only one nonzero solution of $LAI_s$. To obtain the nonzero solution, the numerical approach starts with a guess value of $LAI_s$ and then then iterates to obtain the approximated solution of $LAI_s$ until converging. Note that the numerical approach is widely used in the land surface models. For example, as the stomatal resistance, the CO2 partial pressure at the leaf surface, the internal leaf CO2 partial pressure, and the leaf net photosynthesis are dependent on each other, the Community Land Model 4.5 uses the numerical approach to solve stomatal resistance and leaf photosynthesis iteratively until the internal leaf CO2 partial pressure converges.

P17 L9-11: Not sure what this means? And give reference.

**Reply:** Thank you for your comments. We removed the sentence.

P19 L10-12: This wasn't mentioned earlier. Perhaps in section 2.4 you can mention this.

**Reply:** Thank you for your advice. We did not know that "The SOS and EOS derived from the GLASS data have much larger variation in 1982-2000 than in 2001-2010" until we obtained the results. We therefore choose to keep the sentence where it is.

P21 L1: suggested changes to texts

**Reply:** Thank you for pointing out the typo. We revised the sentence as "Figure 7 compares the simulated GPP using the MOD17 algorithm and LAI derived from different approaches."

P21 L12: SGPD-TS LAI?

**Reply:** Thank you for pointing out the typo. We replaced "MODIS LAI" with "SGPD-TS LAI". It now reads as "Note that the MOD17 algorithm has positive MBE of 0.247 gC/m2/day and 0.571 gC/m2/day when using SGPD-TS LAI and GSI-TS LAI, respectively, as model input data in the site-scale study."

P22 L1: The Xin et al 2018 approach should have been introduced properly in the beginning.
Reply: we agree to introduce the approach better. We revised the introduction as follows:
To allow for predicting the entire LAI time series over a growing season, (Xin et al., 2018) further improved the GPD model by proposing a linear function between LAI and GPP at the steady state. The proposed function and the sophisticated canopy GPP model (i.e., modeling GPP as a function of LAI and climate variables) together form a closed system of equations that includes both vegetation GPP and LAI. The improved GPD model uses the numerical approach, a method that gives an initial value and then iterates to the convergence of the solution, to solve the closed system of equations and derives LAI in the steady state. The improved GPD model then applies the simple moving average method to the steady-state LAI to obtain the modeled LAI time series. The improved method circumvents the need to empirically prescribe a fixed canopy and enables modeling of LAI time series in addition to the timing of individual phenophases.

P22 L10-11: Several existing studies model LAI & GPP prognostically.
**Reply:** we agree with your comments and we removed the sentence that is not appropriate.

P22 17-18: "our modeling approach is essentially based on an ecological assumption" is not sure.
**Reply:** Thank you for your comment. We removed the sentence.

P23 L1-2: What about CO2? GPP also depends on CO2?
**Reply:** Thank you for your asking. We added sentences to discuss the model limitations as folows: "The MOD17 algorithm only assumes the monotonic relationship between air temperature and photosynthesis and between vapor pressure deficit and photosynthesis. It also does not account for the impacts of CO2 on photosynthesis. The use of the MOD17 algorithm in this study thus has limitations in the model structure."

P23 L1-2: How will one find *m* for all PFTs?
**Reply:** Thank you for asking. We added sentences in the discussion chapter as follows: "Another issue is to find the appropriate values of m for different biomes. One way to determine the values of m is to find the regression slope between leaf area index and gross primary production on a monthly basis. Model parameterization however still requires broad tests."

P23 L7: suggested changes to texts
**Reply:** We replaced "via only climate variables" with "forced only by climate variables". It now reads as "To better understand the performance of the developed method, one study is now undertaken to implement the developed method with the Common Land Model for simulating multi-decadal LAI and GPP for global biomes forced only by climate variables."

P23 L20: suggested changes to texts

**Reply:** Thank you for your comments. We deleted "Robust" and revised the sentence as "Terrestrial biosphere models provide a basic tool for understanding the interactions between the land surface and the atmosphere."

P23 L23-24: How does a linear relationship between LAI & GPP support this?

**Reply:** Thank you for your comments. We removed the words that are not appropriate.

P23 L25: This wasn't shown explicitly.

**Reply:** Thank you for your comments. When GPP and LAI are dependent on each other, the solutions can be obtained using the numerical approach. To address your concerns, we revised the texts and added more explanations as follows:

Given the environmental conditions, Equations 1 and 2 together form simultaneous equations, meaning that there are two unknown variables (i.e., LAI and GPP at the steady state) and two different general equations. One may derive an analytic solution if both equations have simple forms. But because the dependence of GPP on LAI is non-linear, deriving the analytic solution is complicated and we could apply the numerical approach to obtain the solutions. Because $LAI_s$ increases as a linear function of $GPP_s$ in Equation 1 and $GPP_s$ increases as a logarithmic function of $LAI_s$ in Equation 2, the simultaneous equations have one and only one nonzero solution of $LAI_s$. To obtain the nonzero solution, the numerical approach starts with a guess value of $LAI_s$ and then then iterates to obtain the approximated solution of $LAI_s$ until converging. Note that the numerical approach is widely used in the land surface models. For example, as the stomatal resistance, the CO2 partial pressure at the leaf surface, the internal leaf CO2 partial pressure, and the leaf net photosynthesis are dependent on each other, the Community Land Model 4.5 uses the numerical approach to solve stomatal resistance and leaf photosynthesis iteratively until the internal leaf CO2 partial pressure converges.

**Once again, thank you for your valuable advice. It really helps us to improve our manuscript.**

[revised manuscript text omitted]

---

## Referee Report (RR1)

*[Handwritten annotations at top:]*
*→ Remove the comparison with moving average scheme.*
*→ This scheme can't be used in LSMs because m = LAI/GPP in not a function of CO₂*

**A simple time-stepping scheme to simulate leaf area index, phenology, and gross primary production across deciduous broadleaf forests in eastern United States**

Qinchuan Xin [1], Yongjiu Dai [2], Xiaoping Liu [1]

[1]Guangdong Key Laboratory for Urbanization and Geo-simulation, Sun Yat-sen University, Guangzhou 510275, China
[2]School of Atmospheric Sciences, Sun Yat-sen University, Guangzhou 510275, China

*Correspondence to*: Qinchuan Xin (xinqinchuan@gmail.com); Yongjiu Dai (daiyj6@mail.sysu.edu.cn)

**Abstract.** Terrestrial plants play a key role in regulating the exchange of energy and materials between the land surface and the atmosphere. Terrestrial biosphere models that simulate leaf dynamics and canopy photosynthesis are required to understand the vegetation-climate interactions. This study proposes a simple time stepping scheme to simulate leaf area index (LAI), phenology, and gross primary production (GPP) when forced with climate variables. The method establishes a linear function between the steady-state LAI and the corresponding GPP, which is used to track the suitability of *[margin note: not sure what this means]* environmental conditions for plant photosynthesis. The method applies the established function and the MOD17 algorithm to *[an]* form simultaneous equations together, which can be solved numerically. To account for the time-lagged responses of plant growth to environmental conditions, a time stepping scheme is developed to simulate the LAI time series based on the solved steady-state LAI. The simulated LAI time series is then used to derive the timing of key phenophases and simulate canopy GPP with the MOD17 algorithm. The developed method is applied to deciduous broadleaf forests in eastern United States and has *[is]* found to perform well on *[for]* simulating canopy LAI and GPP at the site scale as evaluated using both flux tower and satellite data. The method could also captures the spatiotemporal variation of vegetation LAI and phenology across eastern United States as compared with satellite observations. The developed time-stepping scheme provides a simplified and improved version of our previous modeling approach and *[can be]* forms a potential basis for *[applied at]* regional to global applications *[scales]* in future studies. *[↓ to simulate leaf phenology]*

[revised manuscript text omitted]

*Arora & Boer (2005), Global Change Biology, used a similar approach.*

*cannot be used within the framework of*

*This is not a limitation because almost all of these processes are already in land surface models.*

that commonly operate at incremental time steps. Second, the developed GPD model that includes many subtle vegetation processes, such as canopy radiative transfer, leaf stomatal conductance, leaf transpiration, leaf photosynthesis, and soil evaporation, is computationally intensive and requires various climate input data.

Aiming to solve the above-mentioned problems, the objectives of the study are to: 1) develop a time stepping scheme to simulate leaf dynamics and vegetation productivity, and 2) simplify the GPD model to allow for long-term applications at a large scale. Given that the phenology modeling in deciduous broadleaf forest, a biome that have distinct seasonal growing cycles, still has large uncertainties (Melaas et al., 2016), this study choose to simulate the *leaf dynamics for* deciduous broadleaf forests across the eastern United States  successful could provide  future applications to other biomes. *Such a method can be potentially used for*

**2 Methods and materials**

**2.1 Modeling steady-state leaf area index**

One difficulty in vegetation phenology modeling is that the time scale associated with leaf allocation far exceeds that of many other vegetation processes. Unlike leaf photosynthesis that approaches equilibrium within a minute and stomatal functioning that reaches the steady state in minutes (Sellers et al., 1996a), leaf dynamics takes days or even months in response to weather variation (Zeng et al., 2013). (Xin et al., 2018) first put forward the concept of the steady-state leaf area index, i.e., canopy LAI when time approaches infinity while the environmental conditions remain unchanging. An alternative biological explanation to the steady-state LAI is the maximum canopy LAI that an environment can sustain infinitely by its own photosynthetic activities. Supposing that the carrying capacity of canopy LAI is proportional to total canopy photosynthetic rate under a given environment, the steady-state LAI can be modeled as follows:

$$LAI_s = mGPP_s \tag{1}$$

where $LAI_s$ denotes the steady-state leaf area index; m denotes the constant ratio of steady state leaf area index to environmental capacity, *which is* and $GPP_s$  *denoted by* the steady-state gross primary production.

The above equation, despite having a simple form, provides a critical function that complements the canopy photosynthesis model. The only parameter m is dependent on plant functional type and can be quantified from field measurements as the average ratio of LAI to GPP at canopy closure (i.e., the time when both canopy LAI and GPP reach equilibrium). Studies have developed various canopy photosynthesis models, such as the light use efficiency models and the process-based models. Our previous studies (Xin, 2016; Xin et al., 2018) implemented a sophisticated canopy model that assembles the sub-models of canopy radiative transfer, leaf stomatal conductance, leaf transpiration, soil evaporation, and leaf photosynthesis. Although the method has been successfully applied to different biomes, the model structure is complicated for studies at the regional to global scales. To simulate canopy photosynthesis, this study implements the MOD17 algorithm, a big-leaf light use efficiency model that uses routine satellite products (Running et al., 2004). The use of the MOD17 algorithm could greatly simplify the modeling processes and reduce the required climate variables, thereby allowing for broad applications. A brief description on the MOD17 algorithm is provided here where details can be found from the user guide of the MODIS GPP product (Running and Zhao, 2015).

Based on the MOD17 algorithm, vegetation GPP can be modeled as follows:

$$GPP_s = PAR \times FPAR \times \varepsilon_{max} \times f(T) \times f(VPD) \tag{2}$$

where $GPP_s$ denotes the steady-state gross primary production; PAR denotes photosynthetically active radiation; FPAR denotes the fraction of photosynthetically active radiation; $\varepsilon_{max}$ denotes maximum light use efficiency; and $f(T)$ and

$f(VPD)$ denote the scalar functions that account for the limitation of temperature and vapor pressure deficit, respectively, on canopy photosynthesis.

The fraction of photosynthetically active radiation can be modeled as follows (Turner et al., 2006):

$$FPAR = 1 - \exp(-kLAI_s) \tag{3}$$

where k denotes the canopy light extinction coefficient and $LAI_s$ denotes the steady-state leaf area index.

The environmental scalars can be modeled as follows:

$$f(T) = \max\left(\min\left(\frac{TMIN - TMIN_{min}}{TMIN_{max} - TMIN_{min}}, 1\right), 0\right) \tag{4}$$

$$f(VPD) = \max\left(\min\left(1 - \frac{VPD - VPD_{min}}{VPD_{max} - VPD_{min}}, 1\right), 0\right) \tag{5}$$

where TMIN denotes daily minimum air temperature; $TMIN_{min}$ and $TMIN_{max}$ denote the lower and upper thresholds of daily minimum air temperature for vegetation photosynthetic activities, respectively; VPD denotes daily vapor pressure deficit; and $VPD_{min}$ and $VPD_{max}$ denote the lower and upper thresholds of daily vapor pressure deficit for vegetation photosynthetic activities, respectively.

Given the environmental conditions, Equations 1 and 2 together form simultaneous equations, meaning that there are two unknown variables (i.e., LAI and GPP at the steady state) and two different general equations. One may derive an analytic solution if both equations have simple forms. But because the dependence of GPP on LAI is non-linear, deriving the analytic solution is complicated and we could apply the numerical approach to obtain the solutions. Because $LAI_s$ increases as a linear function of $GPP_s$ in Equation 1 and $GPP_s$ increases as a logarithmic function of $LAI_s$ in Equation 2, the simultaneous

I'm hoping later in the manuscript, I'll be told what PAR, T, and VPD is used to calculate LAIs. PAR, T, and VPD vary throughout the year. Is LAIs calculated every time step?

*[handwritten note: non-zero]*

equations have one and only one nonzero solution of $LAI_s$. To obtain the  solution, the numerical approach starts with a guess value of $LAI_s$ and then then iterates to obtain the approximated solution of $LAI_s$ until converging. Note that the numerical approach is widely used in the land surface models. For example, as the stomatal resistance, the CO2 partial pressure at the leaf surface, the internal leaf CO2 partial pressure, and the leaf net photosynthesis are dependent on each other, the Community Land Model 4.5 uses the numerical approach to solve stomatal resistance and leaf photosynthesis iteratively until the internal leaf CO2 partial pressure converges.

**2.2 Modeling leaf area index, phenology, and gross primary production**

Because the physiological processes that plants allocate photosynthates to leaves do not respond instantaneously to climate variation, there is a need to simulate vegetation LAI as lagging behind the steady state. One method to account for the time lagging effect is to apply the simple moving average method to buffer abrupt changes from individual events in the time series. Our previous study applied the simple moving average method to model LAI as the unweighted mean of the previous $LAI_s$ as follows (Xin et al., 2018):

*[handwritten note: I'm still unsure if $LAI_s$ is calculated daily and is a function of time.]*

*[handwritten note: $LAI(n_{day})$]*

$$LAI = \frac{1}{n_{day}} \sum_{i=0}^{n_{day}-1} LAI_s \,(i) \tag{6}$$

where LAI denotes leaf area index at the n day; $n_{day}$ denotes the number of days; i denotes an index starting from 0 to $n_{day} - 1$; and $LAI_s$ denotes the steady state leaf area index.

The simple moving average method, while showing useful in vegetation phenology modeling, is suitable for retrospective analysis rather than prediction, and importantly, it does not match with most land surface models that operate at incremental time steps. Analogous to the method used to simulate leaf stomatal conductance in response to environmental variation, this study proposes a time stepping scheme to simulate LAI realistically as lagging behind the steady state by a simple restricted growth model (Sellers et al., 1996a) as follows:

*[handwritten note: no it just reflects that photosynthesis doesn't instantaneously leads to new/big leaves]*

$$\frac{dLAI}{dt} = k_l(LAI_s - LAI) \tag{7}$$

[revised manuscript text omitted]

LAI can be modeled as the simple moving average of the instantaneous GSI scaled using maximum LAI as follows:

$$GSI = \frac{1}{n_{day}} \sum_{i=0}^{n_{day}-1} iGSI \tag{12}$$

$$LAI = GSI \times LAI_{max} \tag{13}$$

where GSI denotes growing season index at the n day; $n_{day}$ denotes the number of days; i denotes an index starting from 0 to the previous one day; iGSI denotes the instantaneous growing season index; and $LAI_{max}$ denotes the maximum leaf area index at canopy closure.

It is noteworthy that the instantaneous GSI uses the product of the scalars of minimum temperature, vapor pressure deficit, and photoperiod as an indicator to track the potential canopy photosynthetic capacities on the daily basis. Both the GSI model and the SGPD model, despite having different forms, share the same modeling idea. To understand the differences between the simple moving average method and the time stepping method, the GSI model is also implemented with the simple restricted growth model as follows:

$$LAI_s = iGSI \times LAI_{max} \tag{14}$$

$$\frac{dLAI}{dt} = k_l(LAI_s - LAI) \tag{15}$$

where iGSI denotes the instantaneous growing season index; $LAI_{max}$ denotes the maximum leaf area index at canopy closure; $k_l$ denotes a time constant that accounts for the lagged responses of plant leaf allocation to climate variation; and LAI and $LAI_s$ denote the leaf area index and the steady state leaf area index, respectively.

$m = \dfrac{LAI}{GPP_s}$ is likely going to change as atmos $CO_2$ increases. This makes this approach some what not suitable for LSMs which need to be run for future scenarios.

[revised manuscript text omitted]

parameterized across biomes and requires quite limited model inputs of climate variables. Successful implementation with   *LSMs*
the MOD17 algorithm allows for extending the developed method to applications across biomes at regional to global scales.   *for use in earth system models.*

Land surface models that predict vegetation GPP require either satellite-derived LAI input data or the phenology sub-model. The main idea for this study is to improve the phenology modeling by providing time series of LAI simulated using climate variables, and hence enables to simulate GPP forced only by climate variables. Because we implement the MOD17 algorithm instead of the sophisticated process-based model for the purpose of simplicity, one should not expect that GPP simulated based on the model-simulated LAI could be more accurate than GPP simulated based on the satellite-derived LAI.

*→ Yes but not using MOD17 GPP and what about your LAI is not a function of $CO_2$*

The time stepping scheme developed here is also an improvement over the simple moving average method as used in our previous studies. The results obtained using the time stepping method are consistent with the simple moving average method at the site scale and show to be reasonable at the regional scale. Compared to the simple moving average method, the time stepping method could fit seamlessly into the land surface models that operate at incremental time steps such as the Community Land Model and the Common Land Model (Dai et al., 2003). Because the state-of-the-art land surface models all include the canopy photosynthesis sub-model, the developed method can then be easily embedded into these land surface models as an alternative phenology model. Compared to the simple light use efficiency model like the MOD17 algorithm, implementation of the developed time-stepping scheme in the land surface models relies on supercomputing for global applications. To better understand the performance of the developed method, one study is now undertaken to implement the developed method with the Common Land Model for simulating multi-decadal LAI and GPP for global biomes forced only by climate variables.

*to mention explicitly*

*You are forgetting that other than LAI, GPP is a function of a range of climate variables. So improvements in LAI do not lead to same amount of improvement in GPP.*

Applying the developed method to other biomes and other regions still has issues to be solved appropriately. The time stepping method uses the parameter $k_l$ to account for the time lags of leaf allocation in response to environmental changes. For the deciduous broadleaf forests, a biome with strong seasonality, the developed scheme achieved reasonable results with appropriate parameterization. Short vegetation like grasslands tends to respond much quickly to abrupt environment changes like precipitation and tropical ecosystems have strong resilience to short-term environmental variation (Levine et al., 2016; Shen et al., 2011). Another issue is to find the appropriate values of m for different biomes. One way to determine the values of m is to find the regression slope between leaf area index and gross primary production on a monthly basis. Model parameterization however still requires broad tests. These understandings from the observational studies imply that biomes have varied response speeds to the environment and proper model calibration and assessment are required for the developed method. Using the observation data from remote sensing alone is inadequate for model development as satellite-derived LAI could have large uncertainties for some specific biomes other than deciduous broadleaf forests. Fortunately, global flux tower network and regional phenology observation networks are now established and offer abundant data for comprehensive model assessment.

**5 Conclusions**

Terrestrial biosphere models provide a basic tool for understanding the interactions between the land surface and the atmosphere. To provide a complete solution to the simulation of plant leaf dynamics and canopy photosynthesis, this study establishes a linear relationship between the steady state leaf area index and the corresponding canopy photosynthetic capacity. The proposed leaf allocation function complements the canopy photosynthesis model of the MOD17 algorithm to form simultaneous equations that can be solved using the numerical approach. To account for the time lagging of plant leaf allocation in response to climate variation, a time stepping scheme based on a simple restricted growth model is applied to the solved steady state leaf area index to obtain time series of leaf area index. The developed method could perform reasonably well on simulating leaf area index, phenology, and gross primary production for deciduous broadleaf forests across eastern United States over years as found in both the site-scale and regional-scale modeling studies. Compared to the simple moving average method, the time stepping scheme developed here is consistent with and can be easily embedded into the state-of-the-art land surface models that typically operate at incremental time steps. The developed method allows for simulating leaf area index and gross primary production simultaneously and provides a much simplified and improved version of our previous model as a basis for global applications in future studies.

---

## Author Response (AR2)

Dear reviewer,

Thank you for taking time to review our manuscript.
We studied your comments and revised our draft accordingly.
We also went through our manuscript and made corrections on the texts.
Changes to the texts were marked in the manuscript and were presented as follows.
Hope that our revised draft will meet with your approval.

Best wishes,
Qinchuan Xin

**Anonymous Referee # 2**

I appreciate authors' efforts in revising their manuscript. However, I am sorry but I still find that the manuscript falls short of being ready for publication. I have new concerns after reading the manuscript and I summarize my primary concerns below. I am also attaching an annotated version of the manuscript in which authors can find other minor comments.
Reply: please find detailed responses to your annotated comments after our replies to your primary concerns.

At a number of places in the manuscript, the authors have stressed that their scheme can be used for modeling leaf phenology in land surface models like the community land model (CLM). However, one thing that is being neglected here is that models like CLM are able to simulate leaf area index as a function of atmospheric CO2 concentration. In authors' approach the variable m=LAI/GPP is a constant and calculated using present-day LAI and GPP. As such then, it is not possible to use this approach in land surface models that are designed for application for future climate and CO2 scenarios.
Reply: First, our approach is different from others in terms of the leaf phenology sub-model. Although our approach does not account for CO2 directly, it does account for CO2 indirectly if replacing the MOD17 approach with a canopy photosynthesis sub-model that accounts for CO2. If the canopy photosynthesis sub-model accounts for the impacts of CO2, the solved steady-state LAI would become a function of CO2 and so does the modeled LAI. Whether our approach of leaf phenology modeling could account for the impacts of specific climate variables is largely dependent on the GPP models we used.

Second, for the current phenology models, there are nearly no direct considerations for CO2. The table 2 from Richardson et al. (2005) in Global Change Biology summarized the phenology models in 14 terrestrial biosphere model. Most of the them only account for the influence of temperature and none of them accounts for the influences of CO2 directly. In the community land model (CLM), there are also no functions that directly links leaf phenology to CO2. In the phenology sub-model, CLM divided vegetation into 3 classes of the evergreen type, the seasonal-deciduous type, and the

stress-deciduous type. None of these 3 classes include considerations of leaf area index as a function of CO2 directly. It is only able to simulate leaf area index as a function of atmospheric CO2 concentration indirectly.

**Table 2** Summary of models used in this analysis and their representation of phenology and seasonality of leaf area index (LAI). For models with 'prognostic' phenology, the seasonality of LAI is predicted based on climatic drivers; for those with 'prescribed' phenology, an average seasonal LAI cycle, as derived on a site-by-site basis from satellite (AVHRR) data, was used. Models with semi-prescribed and semi-prognostic phenology represent a hybrid of these approaches. GDD is growing degree days; T is temperature; C is carbon; PFT is plant functional type

| Model name | Resolution | Leaf onset | Control on LAI | Leaf loss | Source |
|---|---|---|---|---|---|
| BEPS | Daily | Satellite | Satellite | Satellite | Ju *et al.* (2006) |
| Biome-BGC | Daily | GDD and radiation sum | Dynamic C allocation | Daylength and low temperature | Thornton *et al.* (2002) |
| Can-IBIS | Half-hourly | T threshold | GDD and dynamic C | Prescribed | El Maayar *et al.* (2002) |
| CN-CLASS | Half-hourly | C balance | C balance | Daylength and low temperature | Arain *et al.* (2006) |
| DLEM | Daily | $T_{7\text{-day}}$ > threshold | GDD to PFT limit | Daylength and low temperature | Tian *et al.* (2010) |
| Ecosys | Hourly | Hours above T threshold | Dynamic C allocation | Hours below T threshold | Grant *et al.* (2009) |
| ED2 | Half-hourly | Semi-prescribed | Dynamic C allocation | GDD and leaf turnover | Medvigy *et al.* (2009) |
| ISAM | Half-hourly | Prescribed | Prescribed | Prescribed | Jain & Yang (2005) |
| LoTEC | Half-hourly | GDD | GDD | T-dependent turnover | Hanson *et al.* (2004) |
| LPJ_wsl | Daily | GDD | GDD | Leaf longevity (prescribed) | Sitch *et al.* (2003) |
| ORCHIDEE | Half-hourly | GDD and chilling | Dynamic C allocation | Decreasing T and T threshold | Krinner *et al.* (2005) |
| SiB3 | Half-hourly | Prescribed | Prescribed | Prescribed | Baker *et al.* (2008) |
| SiBCASA | 10 min | Prescribed | Prescribed | Prescribed | Schaefer *et al.* (2008) |
| SSiB2 | Half-hourly | Prescribed | Prescribed | Prescribed | Zhan *et al.* (2003) |

Third, we agree that we need to be cautious about the statements for the model application in land surface models. We removed those inappropriate statements.

I am uncomfortable with authors' comparison of their results with those obtained using the moving average model. The moving average model cannot clearly be used in current generation of land surface models which determine leaf area index prognostically. I suspect the reason the authors have compared results from their new time stepping method to those obtained using the moving average method because this is what they did in the past and are more comfortable with their moving average method. However, this does not mean that the results from the moving average method should be used as a benchmark.

Reply: We believe that making comparisons between the time-stepping method and the moving average method could add values to the manuscript. Second, such comparisons show that the time-stepping method is an alternative way to account for the lagged responses of plants to climate variation. And to our knowledge, this has not been shown before in previous studies. Second, the Growing Season Index, which is based on the moving average method, is widely used in many studies and applications. Third, we do not try to use the moving average method as a benchmark. We use the Growing Season Index as a benchmark but there is a need to show that the time-stepping method can be similarly applied to the Growing Season Index approach such that the two methods can be compared directly.

The justification for developing a simple model is still not strong enough in the revised manuscript.

Reply: First, one purpose of this study were to develop a method that can be used within a time-stepping model framework.

Second, improving the modeling accuracies on leaf phenology is also important. We added the citation to Richardson's publication in Global Change Biology in 2005 as follows "evaluation on 14 land surface models in deciduous forests suggested that almost all models predicted start of the season earlier or end of the season later than observations and the model biases were typically 14 weeks or more". Note that the biases for our modeled results (the MBE values in Table 2) are less than 6 days.

Third, we believe developing a simple model is necessary for many applications. To date, it is still nearly impossible for complex land surface models to operate at fine spatial resolution at global scales because of the computational costs. In our preliminary experiments, it takes nearly two days to run the Common Land model version 2014 for only one modeling year at 1 km solution in Tianhe supercomputer (the second fastest supercomputer in the world). A simple model that could be easily operate at regional to global scales would obviously help us understanding the land surface processes. For the same reasons, although there are complex land surface models, the 1 km MODIS GPP product generated based on the simple MOD17 approach is still popular and has been widely used.

In Figure 6 something doesn't seem right with the GLASS LAI data. Notice that in all panels (a, b, and c) the interannual variability in SOS, EOS, and GSL is much larger before the year 2000.

Reply: We do not think there are anything wrong. Please note that the GLASS data were produced using two different satellite data. From 1982 to 1999, the LAI product was generated at the resolution of $0.05\,°$ from the AVHRR reflectance. From 2000 to 2014, the LAI product was derived from MODIS land surface reflectance (MOD09A1) at the 1 km resolution. It is not surprising that the interannual variability in GLASS reduce significantly after 2000 for the derived SOS, EOS, and GSL.

We added explanations in the texts as follows "Note that the 8-day GLASS LAI product was generated at the $0.05\,°$ resolution using the AVHRR data from 1982 to 1999 and at the 1000 m resolution using the MODIS from 2000 to 2012. The significantly reduced interannual variability for SOS, EOS, and GSL after 2000 in the GLASS data suggests that the use of the AVHRR and MODIS data in the GLASS dataset could contribute uncertainties in the satellite-derived phenological metrics."

In summary, this manuscript needs to be written in such a way so that its primary argument (that a simple prognostic LAI modeling scheme can be used in complex land surface models) can be brought out much more clearly. In its current form, this message is not brought out properly and leaves a reader confused. Perhaps, another reviewer is needed who can look at the manuscript with a fresh set of eyes.

Reply: We agree that we need to be cautious about the argument that the developed modeling scheme can be used in complex land surface models. We removed these statements in the manuscript and only made a few discussions for potential applications.

Page 1 Line 1: Remove the comparison with moving average scheme.

Reply: We believe that making comparisons between the time-stepping method and the moving average method could add values to the manuscript. Second, such comparisons show that the time-stepping method is an alternative way to account for the lagged responses of plants to climate variation. And to our knowledge, this has not been shown before in previous studies. Second, the Growing Season Index, which is based on the moving average method, is widely used in many studies and applications. Third, we do not try to use the moving average method as a benchmark. We use the Growing Season Index as a benchmark but there is a need to show that the time-stepping method can be similarly applied to the Growing Season Index approach such that the two methods can be compared directly.

Page 1 Line 1: This scheme cannot be used in LSMs because m=LAI/GPP is not a function of CO2.

Reply: First, our approach is different from others in terms of the leaf phenology sub-model. Although our approach does not account for $CO_2$ directly, it does account for $CO_2$ indirectly if the MOD17 approach is replaced with a canopy photosynthesis sub-model that account for $CO_2$. If the canopy photosynthesis sub-model accounts for the impacts of $CO_2$, the solved steady-state LAI would become a function of $CO_2$ and so does the modeled LAI. Whether our approach of leaf phenology modeling could account for the impacts of specific climate variables is largely dependent on the GPP models we used. In other words, for land surface models that include the canopy photosynthesis sub-models, the developed method can be embedded into these models as an alternative phenology model if replacing the MOD17 approach with the canopy photosynthesis model.

Second, for the current phenology models, there are nearly no direct considerations for $CO_2$. The table 2 from Richardson et al. (2005) in Global Change Biology summarized the phenology models in 14 terrestrial biosphere model. Most of the them only account for the influence of temperature and none of them accounts for the influences of $CO_2$ directly. In the community land model (CLM), there are also no functions that directly links leaf phenology to $CO_2$. In the phenology sub-model, CLM divided vegetation into 3 classes of the evergreen type, the seasonal-deciduous type, and the stress-deciduous type. None of these 3 classes include considerations of leaf area index as a function of $CO_2$ directly. It is only able to simulate leaf area index as a function of atmospheric $CO_2$ concentration indirectly.

**Table 2** Summary of models used in this analysis and their representation of phenology and seasonality of leaf area index (LAI). For models with 'prognostic' phenology, the seasonality of LAI is predicted based on climatic drivers; for those with 'prescribed' phenology, an average seasonal LAI cycle, as derived on a site-by-site basis from satellite (AVHRR) data, was used. Models with semi-prescribed and semi-prognostic phenology represent a hybrid of these approaches. GDD is growing degree days; T is temperature; C is carbon; PFT is plant functional type

| Model name | Resolution | Leaf onset | Control on LAI | Leaf loss | Source |
|---|---|---|---|---|---|
| BEPS | Daily | Satellite | Satellite | Satellite | Ju *et al.* (2006) |
| Biome-BGC | Daily | GDD and radiation sum | Dynamic C allocation | Daylength and low temperature | Thornton *et al.* (2002) |
| Can-IBIS | Half-hourly | T threshold | GDD and dynamic C | Prescribed | El Maayar *et al.* (2002) |
| CN-CLASS | Half-hourly | C balance | C balance | Daylength and low temperature | Arain *et al.* (2006) |
| DLEM | Daily | $T_{7\text{-day}}$ > threshold | GDD to PFT limit | Daylength and low temperature | Tian *et al.* (2010) |
| Ecosys | Hourly | Hours above T threshold | Dynamic C allocation | Hours below T threshold | Grant *et al.* (2009) |
| ED2 | Half-hourly | Semi-prescribed | Dynamic C allocation | GDD and leaf turnover | Medvigy *et al.* (2009) |
| ISAM | Half-hourly | Prescribed | Prescribed | Prescribed | Jain & Yang (2005) |
| LoTEC | Half-hourly | GDD | GDD | T-dependent turnover | Hanson *et al.* (2004) |
| LPJ_wsl | Daily | GDD | GDD | Leaf longevity (prescribed) | Sitch *et al.* (2003) |
| ORCHIDEE | Half-hourly | GDD and chilling | Dynamic C allocation | Decreasing T and T threshold | Krinner *et al.* (2005) |
| SiB3 | Half-hourly | Prescribed | Prescribed | Prescribed | Baker *et al.* (2008) |
| SiBCASA | 10 min | Prescribed | Prescribed | Prescribed | Schaefer *et al.* (2008) |
| SSiB2 | Half-hourly | Prescribed | Prescribed | Prescribed | Zhan *et al.* (2003) |

Third, we agree that we need to be cautious about the statements for the model application in land surface models. We removed those inappropriate statements.

Page 1 Line 12: not sure what this means "which is used to track the suitability of environmental conditions for plant photosynthesis".
Reply: we removed the texts to avoid confusion.

Page 1 Line 18 - 21: suggested changes to texts.
Reply: we agree with your suggestions. The revised texts now read as follows "The developed method is applied to deciduous broadleaf forests in eastern United States and is found to perform well for simulating canopy LAI and GPP at the site scale as evaluated using both flux tower and satellite data. The method also captures the spatiotemporal variation of vegetation LAI and phenology across eastern United States as compared with satellite observations. The developed time-stepping scheme provides a simplified and improved version of our previous modeling approach to simulate leaf phenology and can be potentially applied at regional to global scales in future studies."

Page 1 Line 25 - 26: suggested changes to texts.
Reply: thanks for your suggestion. It now reads as follows "Terrestrial plants play a key role in regulating the exchange of energy and materials (e.g., radiation, heat and moisture, carbon, and trace gas) between the land surface and the atmosphere (Beer et al., 2010; Zhu et al., 2017). The canopy structure and characteristics govern solar radiation interception and absorption (Ni-Meister et al., 2010; Yuan et al., 2013)."

Page 2 Line 11 - 18: long sentence, break into two

Reply: We break the long sentence into two sentences. It now reads as follows:

"These methods include both the light use efficiency models (e.g., the Carnegie-Ames-Stanford Approach (CASA) model (Potter et al., 1993), the MOD17 algorithm (Running et al., 2004), the Vegetation Photosynthesis Model (VPM) (Xiao et al., 2004), the eddy covariance light use efficiency (EC-LUE) model (Yuan et al., 2010), and the two-leaf light use efficiency (TL-LUE) model (He et al., 2013)) and the process-based models (e.g., the boreal ecosystem productivity simulator (BEPS) model (Liu et al., 1997), the Breathing Earth System Simulator (BESS) model (Ryu et al., 2011), the Growing Production-Day (GPD) model (Xin, 2016), the revised Simple Biosphere (SiB2) model (Sellers et al., 1996b)). Despite different from each other on the representation of vegetation processes, these methods have been successfully used for applications from field to global scales."

Page 2 Line 19 - 20: suggested changes to texts.

Reply: We revised the text as follows "While remotely sensed vegetation data perfectly complements the canopy process models, the ability to dynamically simulate vegetation LAI is fundamental to enhance our abilities on predicting terrestrial ecosystem processes."

Page 3 Line 5 - 7: suggested changes to texts. Weak.

Reply: We agreed that the argument needed improvements and revised the texts as follows "While these studies have greatly benefitted the development of the leaf phenology models, evaluation on 14 land surface models in deciduous forests suggested that almost all models predicted start of the season earlier or end of the season later than observations and the model biases were typically 14 weeks or more. It is therefore necessary to improve the current phenology models."

Page 3 Line 10 - 11: suggested changes to texts.

Reply: Based on your suggestion, we revised the texts as follows "The physiological processes of leaf phenology and canopy photosynthesis are interrelated. Plants absorb carbon dioxide to accumulate biomass through photosynthesis and then redistribute the photosynthetic gain to organs such as leaves, roots, and stems to optimize carbon gain."

Page 3 Line 15 - 16: disconnected and vague.

Reply: We now revised the sentence as "In essence, new leaf phenology models may need to account for the processes of canopy photosynthesis more closely and explicitly than the current leaf phenology models."

Page 3 Line 18: Arora & Boer (2005), Global Change Biology, used a similar approach.

Reply: we studied the paper of Arora & Boer (2005) in Global Change Biology and added it to our citations. It reads as follows "(Arora and Boer, 2005) developed a carbon-gain-based scheme that initiates leaf onset when environmental conditions are beneficial for the plant in carbon terms to produce new leaves and initiates leaf offset when environmental conditions are unfavourable and incur carbon losses for plants."

Page 3 Line 27: suggested changes to texts.

Reply: we revised as you suggested.

Page 3 Line 32: suggested changes to texts.

Reply: We revised as you suggested. It now reads "The improved method allows modeling of LAI time series in addition to the timing of individual phenophases."

Page 3 Line 35: suggested changes to texts.

Reply: We revised as you suggested. It now reads "There remain shortcomings to overcome for the broad applications of the GPD model. First, the simple moving average method, despite being widely used in many studies, is empirical and cannot be used within the framework of models that operate at incremental time steps. Second, the developed GPD model that includes many subtle vegetation processes, such as canopy radiative transfer, leaf stomatal conductance, leaf transpiration, leaf photosynthesis, and soil evaporation, is computationally intensive and requires various climate input data for regional to global applications."

Page 4 Line 1: Justification for developing a simple model still not strong enough.

Reply: we agree with your argument. One main purpose of this study were to develop a method that can be used within a time-stepping model framework. Another reason is to improve the modeling accuracies. We therefore added the citation to Richardson's publication in Global Change Biology in 2005 as follows "evaluation on 14 land surface models in deciduous forests suggested that almost all models predicted start of the season earlier or end of the season later than observations and the model biases were typically 14 weeks or more". Note that the biases for our modeled results (the MBE values in Table 2) are less than 6 days.

Page 4 Line 1: This is not a limitation because almost all of these processes are already in land surface models.

Reply: thank you for your comments. We agree that this is not a limitation but the computational costs are expensive at fine spatial resolution for large-scale applications. For example, it is nearly impossible to perform 1 km resolution global simulations using the current land surface models even with supercomputing. We revised the related texts as "is computationally intensive for regional to global applications".

Page 4 Line 8 - 10: suggested changes to texts.

Reply: We revised the texts based on your suggestions. It now reads as follows "this study choose to simulate leaf dynamics for the deciduous broadleaf forests across the eastern United States. If successful, such a method can be potentially used for future applications to other biomes."

Page 4 Line 23: suggested changes to texts.

Reply: We revised the sentence and it now reads as follows "where $LAI_s$ denotes the steady-state leaf area index; m denotes the constant ratio of steady state leaf area index to environmental capacity denoted by $GPP_s$, which is the steady-state gross primary production."

Page 5 Line 1: Does not this precludes the MOD17 approach from being used in prognostic models for future scenarios. This is the justification you gave for developing your LAI model (Line 20, Page 2)

Reply: The MOD17 approach cannot be used in prognostic models for future scenarios because it requires the inputs of satellite-derived LAI data. Our developed method provides the simulated LAI input data for the MOD17 approach such that the MOD17 approach can then be used as prognostic models without observational satellite data.

Page 5 Line 4: suggested changes to texts

Reply: we revised as you suggested. It now reads "A brief description of the MOD17 algorithm is provided here and details can be found from the user guide of the MODIS GPP product (Running and Zhao, 2015)."

Page 5 Line 8: why is this steady state if PAR, T, & VPD are time varying?

Reply: For each day when PAR, T, and VPD are given, the vegetation GPP corresponding to the steady-state leaf area index can be modeled using the MOD17 algorithm. We revised the equations to be more clear.

Based on the MOD17 algorithm, vegetation GPP corresponding to the steady-state leaf area index can be modeled as follows:

$$GPP_s = PAR \times FPAR_s \times \varepsilon_{max} \times f(T) \times f(VPD) \tag{1}$$

where $GPP_s$ denotes the gross primary production corresponding to the steady-state leaf area index; $PAR$ denotes photosynthetically active radiation; $FPAR_s$ denotes the fraction of photosynthetically active radiation corresponding to the steady-state leaf area index; $\varepsilon_{max}$ denotes maximum light use efficiency; and $f(T)$ and $f(VPD)$ denote the scalar functions that account for the limitation of temperature and vapor pressure deficit, respectively, on canopy photosynthesis.

Page 5 Line 24: for given values of PAR, T, & VPD?

Reply: You are correct. We revised the sentence as follows "Given the environmental conditions (i.e., given daily values of photosynthetically active radiation, temperature, and vapor pressure deficit), Equations 1 and 2 together form simultaneous equations, meaning that there are two unknown variables (i.e., LAI and GPP at the steady state) and two different general equations."

Page 5 Line 26: I am hoping later in the manuscript, I will be told what PAR, T, and VPD is used to calculate LAIs. PAR, T, and VPD vary throughout the year. Is LAIs calculated every time step?

Reply: LAIs is calculated every time step on the daily basis. We added a sentence to explain it as follows "For every day, daily photosynthetically active radiation, daily minimum air temperature, and daily vapor pressure deficit are used as the forcing data to calculate $LAI_s$ for the corresponding day."

Page 6 Line 1: suggested changes to texts

Reply: we revised as you suggested. It now reads as follows "To obtain the non-zero solution, the numerical approach starts with a guess value of $LAI_s$ and then then iterates to obtain the approximated solution of $LAI_s$ until converging."

Page 6 Line 12: I am still unsure if LAIs is calculated daily and is a function of time.
Reply: LAIs is calculated every time step on the daily basis. LAIs is not a function of time directly but is a function of PAR, T, and VPD. Because PAR, T, and VPD vary throughout the year, the calculated LAIs vary from day to day.

We added sentences as follows "For every day, daily photosynthetically active radiation, daily minimum air temperature, and daily vapor pressure deficit are used as the forcing data to calculate $LAI_s$ for the corresponding day. Because photosynthetically active radiation, minimum air temperature, and vapor pressure deficit vary throughout the year, the calculated $LAI_s$ vary from day to day."

Page 6 Line 13: suggested changes to the equation.
Reply: we now revised the equation as follows:

$$LAI(n+1) = \frac{1}{n}\sum_{i=1}^{n} LAI_s(i) \qquad (2)$$

where $LAI(n+1)$ denotes leaf area index at the $n+1$ day; $n$ denotes the number of days; $i$ denotes an index starting from 1 to $n$; and $LAI_s$ denotes the steady state leaf area index.

Page 6 Line 22: no it just reflects that photosynthesis does not instantaneously leads to new/big leaves
Reply: we revised as you suggested. It now reads as "$k_l$ denotes a time constant that reflects that photosynthesis does not instantaneously lead to new or big leaves".

Page 7 Line 12: what is the LAI on $1^{st}$ day of spring? You have to use either an imaginary LAI to get GPP started or you need to push leaves out (like in the real world deciduous trees use non-structural carbohydrates from previous years). So I am still unsure how does GPP becomes on the very $1^{st}$ day of spring.
Reply: Our modeling approach is different from your understanding. For the first day of spring, the modeled LAI is zero, the modeled fraction of photosynthetically active radiation (FPAR) is zero, and the modeled GPP is zero. When time moves forward, the modeled steady-state LAI (i.e., $LAI_s(n)$) is non-zero. Based on Equation 7, the modeled increases in LAI (i.e., $\frac{dLAI(n)}{dt}$) is positive. And then in Equation 8, the modeled LAI on the next time step is not zero anymore and starts to increase. To make it clear, we revised the texts and also added Equation 8 as follows.

$$\frac{dLAI(n)}{dt} = k_l[LAI_s(n) - LAI(n)] \qquad (3)$$

$$\text{LAI}(n+1) = \text{LAI}(n) + \frac{d\text{LAI}(n)}{dt} \tag{4}$$

where $t$ denotes the time; $k_l$ denotes a time constant that reflects that photosynthesis does not instantaneously lead to new or big leaves; and $\text{LAI}(n)$ and $\text{LAI}_s(n)$ denote the leaf area index and the steady state leaf area index at the n day, respectively.

Page 7 Line 25: suggested changes to texts
Reply: we revised the texts following your suggestion.

Page 9 Line 1: m=LAI/GPP is likely going to change as atmosphere CO2 increases. This makes this approach somewhat not suitable for LSMs which need to be used for future scenarios.
Reply: It is difficult to know whether the relationship would change or not change as atmosphere CO2 changes. We believe that there are no reasons to add some relationships or mechanisms that we do not know yet into models. One concern you have is that you thought our modeled LAI was not a function of CO2. First, although our approach does not account for CO2 directly, it does account for CO2 indirectly if the MOD17 approach is replaced with a canopy photosynthesis sub-model that account for CO2. If the canopy photosynthesis sub-model accounts for the impacts of CO2, the solved steady-state LAI would become a function of CO2 and so does the modeled LAI. Whether our approach of leaf phenology modeling could account for the impacts of specific climate variables is largely dependent on the GPP models we used. Second, for the current phenology models, there are nearly no direct considerations for CO2. The table 2 from Richardson et al. (2005) in Global Change Biology summarized the phenology models in 14 terrestrial biosphere model. Most of the them only account for the influence of temperature and none of them accounts for the influences of CO2 directly.

**Table 2** Summary of models used in this analysis and their representation of phenology and seasonality of leaf area index (LAI). For models with 'prognostic' phenology, the seasonality of LAI is predicted based on climatic drivers; for those with 'prescribed' phenology, an average seasonal LAI cycle, as derived on a site-by-site basis from satellite (AVHRR) data, was used. Models with semi-prescribed and semi-prognostic phenology represent a hybrid of these approaches. GDD is growing degree days; T is temperature; C is carbon; PFT is plant functional type

| Model name | Resolution | Leaf onset | Control on LAI | Leaf loss | Source |
|---|---|---|---|---|---|
| BEPS | Daily | Satellite | Satellite | Satellite | Ju *et al.* (2006) |
| Biome-BGC | Daily | GDD and radiation sum | Dynamic C allocation | Daylength and low temperature | Thornton *et al.* (2002) |
| Can-IBIS | Half-hourly | T threshold | GDD and dynamic C | Prescribed | El Maayar *et al.* (2002) |
| CN-CLASS | Half-hourly | C balance | C balance | Daylength and low temperature | Arain *et al.* (2006) |
| DLEM | Daily | $T_{7\text{-day}}$ > threshold | GDD to PFT limit | Daylength and low temperature | Tian *et al.* (2010) |
| Ecosys | Hourly | Hours above T threshold | Dynamic C allocation | Hours below T threshold | Grant *et al.* (2009) |
| ED2 | Half-hourly | Semi-prescribed | Dynamic C allocation | GDD and leaf turnover | Medvigy *et al.* (2009) |
| ISAM | Half-hourly | Prescribed | Prescribed | Prescribed | Jain & Yang (2005) |
| LoTEC | Half-hourly | GDD | GDD | T-dependent turnover | Hanson *et al.* (2004) |
| LPJ_wsl | Daily | GDD | GDD | Leaf longevity (prescribed) | Sitch *et al.* (2003) |
| ORCHIDEE | Half-hourly | GDD and chilling | Dynamic C allocation | Decreasing T and T threshold | Krinner *et al.* (2005) |
| SiB3 | Half-hourly | Prescribed | Prescribed | Prescribed | Baker *et al.* (2008) |
| SiBCASA | 10 min | Prescribed | Prescribed | Prescribed | Schaefer *et al.* (2008) |
| SSiB2 | Half-hourly | Prescribed | Prescribed | Prescribed | Zhan *et al.* (2003) |

Page 9 Line 29: suggested changes to texts

Reply: we revised the texts following your suggestion.

Page 14 Line 1: unclear what this means and suggested changes to texts.

Reply: we removed the words that are unclear and revised the texts according to your suggestion. It now reads "The SGPD-based models generally outperform the GSI-based models as the achieved correlation coefficients are higher and the RMSE are smaller. Both the GSI-SMA and GSI-TS models predict spring onsets earlier than observations by more than 30 days and predict autumn senescence later than observations by more than 20 days."

Page 14 Line 11: what does this mean?

Reply: We revised the texts as "The modeled and measured GPP are compared in Figure 3 to understand the performance of GPP modeling."

Page 15 Line 10-11: suggested changes to texts

Reply: we replaced the word "extents" with "distributions" as you suggested.

Page 17 Line 6: suggested changes to texts

Reply: we replaced the word "extents" with "distributions" as you suggested.

Page 20 Line 15: Fig 6 needs more discussion. The R2 between MODIS and GLASS values for period 2000 – 2010 should also be shown and discussed to see how the two observation-based estimates compare to each other. This will be helpful before model can be compared to these

observation-based estimates.

Reply: we polished Fig 6 and added discussion to the results "The correlation coefficients between the GLASS data and the MODIS data for SOS, EOS, and GSL from 2001 to 2014 are 0.892, 0.412, and 0.288, respectively. There are only 14 years overlapping between these two different datasets and the correlations are insignificant for both the derived EOS and GSL."

Page 21 Line 1: Something does not seem right in GLASS data. How come the interannual variability in GLASS reduce significantly after 2000 for SOS, EOS, and GSL? What is this range? Please clarify.

Reply: We do not think there are anything wrong. Please note that the GLASS data were produced using two different satellite data. From 1982 to 1999, the LAI product was generated at the resolution of 0.05 ° from the AVHRR reflectance. From 2000 to 2014, the LAI product was derived from MODIS land surface reflectance (MOD09A1) at the 1 km resolution. It is not surprising that the interannual variability in GLASS reduce significantly after 2000 for the derived SOS, EOS, and GSL.

We added explanations in the texts as follows "Note that the 8-day GLASS LAI product was generated at the 0.05 ° resolution using the AVHRR data from 1982 to 1999 and at the 1000 m resolution using the MODIS from 2000 to 2012. The significantly reduced interannual variability for SOS, EOS, and GSL after 2000 in the GLASS data suggests that the use of the AVHRR and MODIS data in the GLASS dataset could contribute uncertainties in the satellite-derived phenological metrics."

In addition, to explain the range in the figure, we added one sentence to the figure caption as follows "The shaded areas denote the standard deviation of the corresponding phenophases across spaces."

Page 23 Line 11: This sounds like a huge statement. But this gap is already been bridged in many land models.

Reply: We agree with you and revised the sentence as follows "Here we provide a simple time-stepping solution that allow for simulating canopy photosynthesis, leaf area index, and leaf phenology simultaneously."

Page 24 Line 8: Of course, why do you need Fig 8 to tell this?

Reply: thank you for your comments. We revised the sentence as follows "Figure 8 suggests that LAI has stronger correlation with GPP than with temperature on the monthly basis."

Page 24 Line 10: Is this subset of Fig 2a?

Reply: It is not a subset of Fig 2a. The analysis was performed at different time scales. We added a sentence to the figure caption "All data were averaged to the monthly time scale for analysis, making the point numbers different from the analysis at the weekly time scale in Figure 2.."

Page 25 Line 13: suggested changes to texts.

Reply: we revised the sentence based on your suggestions. It now reads "It implies that the LAI modeling in our developed method will likely benefit from improvements on the canopy photosynthesis model."

Page 25 Line 14: Yes, of course, but not in LSMs for use in earth system models.
Reply: In this study we implement with the MOD17 approach to make predictions of LAI and GPP. If we replace the MOD17 approach with the canopy photosynthesis sub-model in the land surface model, the method could be applied to simulate both LAI and GPP as well. We agree that we have to be cautious with our statements before tests on applications in land surface models. We therefore removed the statements that are related to the applications for use in land surface models.

Page 25 Line 28: Yes but not using MOD17 GPP and what about your LAI is not a function of $CO_2$.
Reply: we agree that it should be not used with the MOD17 approach and we revised the texts as "For land surface models that include the canopy photosynthesis sub-models, the developed method can be embedded into these models as an alternative phenology model if replacing the MOD17 approach with the canopy photosynthesis model."

Another concern you have is that the modeled LAI is not a function of $CO_2$. First, although our approach does not account for $CO_2$ directly, it does account for $CO_2$ indirectly if replacing the MOD17 approach with a canopy photosynthesis sub-model that accounts for $CO_2$. If the canopy photosynthesis sub-model accounts for the impacts of $CO_2$, the solved steady-state LAI would become a function of $CO_2$ and so does the modeled LAI. Whether our approach of leaf phenology modeling could account for the impacts of specific climate variables is largely dependent on the GPP models we used. Second, for the current phenology models, there are nearly no direct considerations for $CO_2$. The table 2 from Richardson et al. (2005) in Global Change Biology summarized the phenology models in 14 terrestrial biosphere model. Most of the them only account for the influence of temperature and none of them accounts for the influences of $CO_2$ directly.

Page 25 Line 35: You are forgetting to mention explicitly that other than LAI, GPP is a function of a range of climate variables. So improvements in LAI do not lead to same amount of improvements in GPP.
Reply: We agree with your advice and added a sentence to the texts. It now reads "Because GPP is 
[revised manuscript text omitted]